# CONSTRUCT-THEN-COMPRESS: GEOMETRIC DYNAMICS OF GROKKING IN TRANSFORMERS

## ABSTRACT

A central puzzle in deep learning is how generalized algorithms emerge from training dynamics, particularly in the phenomenon of grokking. Existing approaches track function complexity (Linear Mapping Number) or representation dimensionality (Local Intrinsic Dimension). We take a different perspective: a unified algorithm should manifest as geometrically consistent transformations across inputs. We introduce the **Geometric Coherence Score (GCS)**, which measures the directional alignment of local Jacobian transformations across the data manifold. GCS provides a geometric signature of mechanistic unity—consistent transformations indicate a unified computational strategy, while scattered transformations suggest input-specific memorization. Combined with a fixed final geometry protocol that isolates mechanistic evolution from geometric drift, GCS reveals a **Construct-then-Compress** dynamic—specifically, a reduction in geometric modes rather than representational dimensions—invisible to complexity or dimensionality metrics. In single-layer Transformers, this dynamic unfolds in three distinct phases: (1) *Coherence Collapse*, where initial symmetry breaks to memorize data; (2) *Asynchronous Construction and Compression*, a critical silent phase where Attention initiates geometric reorganization, followed by MLP with temporal offset; and (3) *Post-Grokking Refinement*, where the mechanism consolidates into a unified solution. We validate the construct-then-compress principle across activation functions (ReLU, GeLU, SiLU) and modular tasks (addition, subtraction, multiplication, division), establishing GCS as a principled diagnostic tool. Extending to multi-layer networks (2–3 layers), we observe that final layers exhibit iterative construct-compress cycles rather than a single three-phase trajectory, while early layers show path-specific stability. These findings reveal depth-dependent dynamics that warrant further investigation into how hierarchical structure shapes algorithmic formation.

## 1 INTRODUCTION

Understanding how neural networks transition from memorization to generalization remains a fundamental challenge in deep learning. This question has gained urgency with the rise of Large Language Models, which exhibit emergent abilities that are not explicitly programmed (Havlík, 2025). The phenomenon of Grokking—where a network's generalization performance suddenly spikes long after memorizing the training data—serves as a canonical testbed for investigating this mystery (Power et al., 2022). Observed across models from small transformers to Large Language Models (Liu et al., 2023a; Li et al., 2025; Humayun et al., 2024), this phenomenon challenges our understanding of the memorization-generalization transition. While numerous theories have been proposed—from competing circuits to phase transitions (Merrill et al., 2023; Carvalho et al., 2025; Liu et al., 2022; Rubin et al., 2024; Varma et al., 2023)—the precise mechanism of how a network transitions from brute-force memorization to algorithmic understanding remains elusive.

Recent breakthroughs in mechanistic interpretability offer a powerful new lens for this investigation. In modular addition, researchers have successfully reverse-engineered the final learned algorithm, revealing that trained transformers implement sophisticated solutions based on discrete Fourier transforms (Nanda et al., 2023), clock-like circular representations (Zhong et al., 2023), constructive analytical solutions (Gromov, 2023), or universal abstract algorithms (McCracken et al.,

2025). While we now know what elegant solution the network finds, the fundamental question persists: **how is this algorithm formed?**

To illuminate this process of algorithmic formation, we adopt a geometric perspective grounded in the manifold hypothesis (Cayton et al., 2005; Meilă & Zhang, 2024). Our key insight is that a unified algorithm should manifest as geometrically consistent transformations: if a network has learned a coherent computational strategy, it should transform similar inputs in similar ways. Conversely, scattered, input-specific transformations indicate memorization rather than algorithmic understanding. We introduce the Geometric Coherence Score (GCS), which quantifies the directional alignment of local Jacobian transformations across the data manifold. Unlike metrics that measure function complexity or representation dimensionality, GCS asks: *how consistently does the network transform different inputs?* This geometric consistency serves as a signature of mechanistic unity—providing a high-resolution view of when and how algorithmic structure emerges during training (Figure 1).

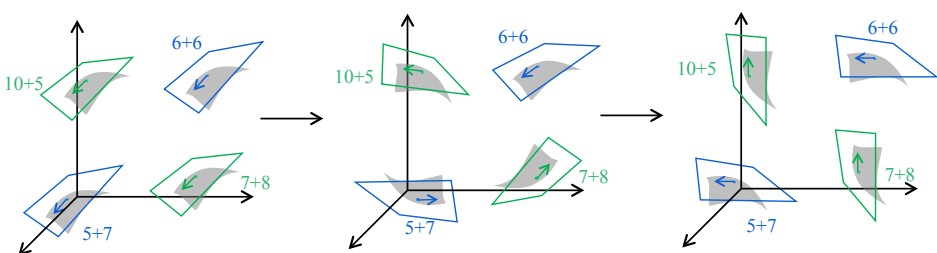

Figure 1: **The Evolution of Geometric Coherence.** Our GCS metric reveals a non-monotonic learning process. (**left**) Trivial Coherence: An initial, non-generalizing state where the network applies a single, simple geometric transformation to all inputs, resulting in high GCS. (**middle**) Complexity Construction: The network learns specialized, inconsistent transformations for different inputs, causing a necessary drop in GCS. (**right**) Emergent Coherence: The network discovers a generalizing solution by unifying transformations for inputs of the same semantic class, while maintaining distinct transformations for different classes. This sophisticated, class-conditional coherence is the hallmark of the grokked state.

Applying this geometric analysis to a single-layer Transformer, we uncover a construct-then-compress dynamic that orchestrates generalization. The network undergoes a non-monotonic, three-stage evolution: **(I) Coherence Collapse:** Initial spurious symmetry breaks as all pathways synchronously decrease coherence to memorize disjoint data points. **(II) Asynchronous Construction and Compression:** While test accuracy remains flat, GCS reveals active structural evolution—Attention initiates geometric compression early by constructing ordered representations that eliminate redundant geometric degrees of freedom, while MLP follows with temporal offset. **(III) Post-Grokking Refinement:** System-wide unification occurs as MLP completes compression and Attention undergoes characteristic double descent, stabilizing into the final coherent algorithm.

We substantiate this discovery as follows:

- **Universality:** The construct-then-compress principle is consistent across diverse activation functions (ReLU, GeLU, SiLU) and modular operations (addition, subtraction, multiplication, division). In multi-layer networks (2–3 layers), final layers exhibit iterative construct-compress cycles, revealing depth-dependent dynamics that warrant further investigation.

- **Falsifiability:** The dynamic is absent in overfitting regimes (Appendix E), confirming that it specifically signifies algorithmic generalization rather than generic training artifacts.

- **Mechanistic Interpretability:** GCS directly tracks the evolution of attention patterns (Appendix G), grounding our geometric measurements in concrete circuit-level changes.

Our work reframes grokking as a process of asynchronous geometric reorganization, offering a principled framework for understanding how generalization emerges from the interplay of hierarchical depth and modular complexity.

## 2 RELATED WORK

**Quantitative Metrics for Grokking.** Existing metrics can be categorized by what they measure: *function complexity*—LMN (Liu et al., 2023b) counts piecewise-linear regions; *representation dimensionality*—LID (Ruppik et al., 2025) and geometric regularizers (Walker et al., 2025) track intrinsic dimension; *transformation magnitude*—Jacobian regularization constrains smoothness. Our work asks a different question: *how consistently does the network transform different inputs?* We propose that transformation consistency serves as a geometric signature of mechanistic unity, shifting focus from descriptive statistics to coherence and revealing dynamics invisible to other metrics.

**Theoretical Mechanisms Proposed for Grokking.** The phenomenon of grokking (Power et al., 2022), where generalization is dramatically delayed, is a canonical example of emergence in deep learning and has been observed in models as large as LLM (Li et al., 2025). The effort to explain this dynamic has produced a rich and diverse landscape of theoretical hypotheses. These include mechanistic theories centered on the discovery of specific algorithms, such as the discrete Fourier transform (Nanda et al., 2023), or the competition between memorizing and generalizing circuits (Varma et al., 2023; Merrill et al., 2023). Other lines of work attribute the phenomenon to the dynamics of optimization, positing it as a phase transition in the loss landscape (Liu et al., 2022) or a consequence of the optimizer's implicit bias (Lv et al., 2025). Although these theories provide valuable high-level perspectives, a key challenge remains to quantitatively track the underlying structural changes in the network function itself.

Recent work has begun to connect grokking to broader phenomena in deep learning. Kumar et al. (2024) frames grokking as a transition from lazy to rich training dynamics, where networks shift from using simple initial features to learning complex, task-specific representations—a perspective further developed by Chou et al. (2025) through representational geometry analysis. Others have identified deep connections to double descent (Davies et al., 2022; Huang et al., 2024), suggesting that grokking, double descent, and circuit competition may arise from unified geometric principles. Complementing these theoretical perspectives, several works have investigated the specific structure of learned algorithms in modular arithmetic tasks (Morwani et al., 2024), revealing how features emerge through implicit regularization. Our work contributes to this landscape by providing the first direct geometric measurements of these proposed dynamics.

## 3 METHOD

Our methodology introduces a novel framework for quantifying the functional complexity of neural networks from a geometric perspective. We begin by establishing the theoretical principles that motivate our approach, then provide a rigorous algorithmic definition of our proposed metric, the Geometric Coherence Score (GCS).

### 3.1 THEORETICAL MOTIVATION: FROM ALGORITHMIC CONSISTENCY TO GEOMETRIC COHERENCE

The central challenge in understanding grokking is quantifying when a network transitions from memorizing individual examples to learning a **unified algorithmic strategy**. Traditional metrics like loss and accuracy capture performance but not the *consistency* of computational strategies across inputs. We propose that this consistency can be measured geometrically. If a network learns a unified algorithm, it should apply similar geometric transformations to the internal representations of different inputs. Conversely, a memorizing network employs disparate, input-specific transformations.

This perspective differs fundamentally from metrics like Participation Ratio (PR), which quantify the *shape* of representations (e.g., effective dimensionality) but not *how* those representations are transformed. PR derives from the covariance of activations, while GCS derives from the consistency of the Jacobian $\mathbf{J}_f$—these are orthogonal properties. A network can reorganize its internal mechanism to be more coherent without changing its representational dimensionality; GCS detects such "iso-dimensional" reorganization phases that purely dimensional metrics miss (see Appendix F).

Our approach is inspired by the principles of Manifold Learning. The **Manifold Hypothesis** posits that high-dimensional data reside on a low-dimensional intrinsic manifold (Meilă & Zhang, 2024).

However, our goal is not to learn a new low-dimensional embedding. Instead, we propose a new paradigm: using the data manifold as a geometric reference frame to analyze the properties of the learned network function $f$. We hypothesize that generalization corresponds to the emergence of **Geometric Coherence**, the degree to which $f$ applies a consistent geometric transformation to local structures (tangent spaces) across the manifold. A high degree of coherence signifies that the network has discovered a simple, universal algorithm that unwraps the manifold's complexity. A low degree of coherence indicates a complex, inconsistent function characteristic of memorization.

## 3.2 QUANTIFYING GEOMETRIC COHERENCE

To make this concept precise, we introduce a multi-step algorithm that translates the abstract notion of "geometric coherence" into a single, quantitative score.

**Local Tangent Space Estimation.** Given a computational flow $f : \mathcal{R}_{\text{in}} \to \mathcal{R}_{\text{out}}$, we construct the reference manifold in the input activation space of a converged reference model $f_{\text{ref}}$. For each sampled input $\mathbf{x}_i$, we extract its internal representation $\mathbf{r}_i \in \mathbb{R}^D$ at the flow's input layer. For Transformers, we extract the residual stream at the **final token position** (the "=" token), which aggregates task-relevant computation.

We estimate the tangent space $T_{\mathbf{r}_i}\mathcal{M}$ by identifying the $k$-nearest neighbors $\mathcal{N}_i$ in the representation space and forming a centered matrix $\mathbf{X}_i \in \mathbb{R}^{k \times D}$ with rows $(\mathbf{r}_j - \mathbf{r}_i)$ for $j \in \mathcal{N}_i$. SVD yields an orthonormal basis $\{\mathbf{v}_{i,1}, \ldots, \mathbf{v}_{i,d}\}$ from the first $d$ right singular vectors. Crucially, the *same* neighborhood $\mathcal{N}_i$ is used both to estimate the tangent space and to define the edges in the coherence matrix $\mathbf{G}$—this ensures that we measure how consistently the network transforms the very geometric structure (the local neighborhood) from which the tangent basis was derived.

SVD provides a *canonical ordering* by variance magnitude ($\sigma_1 \geq \sigma_2 \geq \cdots$): $\mathbf{v}_{i,1}$ is the direction of maximal local variation, $\mathbf{v}_{i,2}$ the second-most, etc. This data-driven ordering is numerically stable in neural networks due to their strong anisotropy (Ethayarajh, 2019)—representations occupy narrow cones rather than uniform spheres, ensuring well-separated singular values (see Appendix D).

**Network Transformation via JVP.** We use the Jacobian-Vector Product (JVP) to compute how each tangent vector is transformed:

$$\mathbf{v}'_{i,\ell} = \mathbf{J}_f(\mathbf{e}_i)\,\mathbf{v}_{i,\ell}, \tag{1}$$

where $\mathbf{J}_f(\mathbf{e}_i)$ is the Jacobian of flow $f$ at the embedding $\mathbf{e}_i$. For Transformers, the tangent vector is embedded into the sequence space with nonzero values only at the final token position, restricting the JVP to measure geometry transformation at the task-critical output position.

**The Geometric Coherence Matrix G.** The core insight is that if a network has learned a coherent algorithm, it should transform the local geometry of neighboring points in similar ways. For each neighbor pair $(i, j)$ with $j \in \mathcal{N}_i$, we measure the alignment of their transformed tangent bases:

$$G_{ij} = \frac{1}{d} \sum_{\ell=1}^{d} \frac{|\langle \mathbf{v}'_{i,\ell}, \mathbf{v}'_{j,\ell} \rangle|}{\|\mathbf{v}'_{i,\ell}\|\|\mathbf{v}'_{j,\ell}\|} \tag{2}$$

Each term compares $\mathbf{v}'_{i,\ell}$ with $\mathbf{v}'_{j,\ell}$—the $\ell$-th transformed tangent vectors from each point. This index-wise correspondence leverages the SVD's canonical ordering: since $\mathbf{v}_{i,1}$ always captures the direction of maximal local variance, comparing $\mathbf{v}'_{i,1}$ with $\mathbf{v}'_{j,1}$ asks whether the network transforms the "most important local direction" consistently across neighbors. For neighboring points on a smooth manifold, these principal directions are naturally aligned, making index-wise comparison geometrically meaningful. This measures whether the network transforms the *same geometric structure* consistently—capturing manifold coherence rather than abstract subspace overlap.

Following standard practice in manifold learning (Tenenbaum et al., 2000; Belkin & Niyogi, 2003), we set $G_{ij} = 0$ for non-neighboring pairs, restricting measurement to coherence *along* the data manifold rather than *across* it. Distant points may have correlated tangents by coincidence, but a unified algorithm should produce consistent transformations specifically for inputs that are locally similar on the learned representation manifold. This local-to-global construction—building global coherence from local consistency—allows the spectral analysis to reveal whether local coherences aggregate into a globally coherent transformation. We set $G_{ii} = 1$ (self-similarity).

**The Geometric Coherence Score (GCS).** To aggregate local coherence into a global score, we analyze the eigenvalue spectrum $\{\lambda_1, \ldots, \lambda_N\}$ of $\mathbf{G}$. We normalize the spectrum as a probability distribution $p_i = |\lambda_i| / \sum_j |\lambda_j|$ and compute its Von Neumann entropy(Petz, 2001):

$$S_{\mathrm{NL}} = -\sum_{i=1}^{N} p_i \log_2 p_i \tag{3}$$

The Geodesic Mode Number (GMN), defined as $\mathrm{GMN} = 2^{S_{\mathrm{NL}}}$, represents the effective number of independent geometric modes.

Finally, we define our primary reported metric, the **Geometric Coherence Score (GCS)**, as:

$$\mathrm{GCS} = N - \mathrm{GMN} \tag{4}$$

A random function yields $\mathrm{GMN} \approx N$ and $\mathrm{GCS} \approx 0$; thus GCS quantifies the reduction from this random baseline—intuitively, the number of geometric modes unified into a coherent algorithm. A complete derivation is in Appendix A. The procedure is summarized in Algorithm 1.

### 3.3 Modular Analysis of Transformer Computational Flows

To analyze complex architectures like Transformers, we apply the GCS metric not only to the entire network but to specific sub-functions, which we term **Computational Flows**. A flow is a well-defined function from an input activation space to an output activation space (e.g., from the block's input to the FFN's output). This modular approach transforms GCS from a global score into a surgical tool for dissecting a network's internal algorithm. In the following sections, we apply this framework to reveal the remarkable learning dynamic of a Transformer undergoing grokking.

---

**Algorithm 1** GCS Computation with Fixed Geometry Protocol

---

**Require:** Checkpoint $f$, reference model $f_{\mathrm{ref}}$, samples $\{\mathbf{x}_i\}_{i=1}^N$, $k$, $d$
**Ensure:** Geometric Coherence Score (GCS)
  1: **Build fixed geometry from $f_{\mathrm{ref}}$:**
  2:   Extract $\mathbf{r}_i \leftarrow f_{\mathrm{ref}}(\mathbf{x}_i)[-1, :], \quad \mathbf{e}_i \leftarrow \mathrm{Embed}_{f_{\mathrm{ref}}}(\mathbf{x}_i)$ {Final token}
  3: **for** $i = 1$ to $N$ **do**
  4:   Find $k$-NN $\mathcal{N}_i$; SVD on centered neighbors $\rightarrow$ tangent basis $\{\mathbf{v}_{i,\ell}\}_{\ell=1}^d$
  5: **end for**
  6: **Analyze checkpoint $f$:**
  7: **for** $i = 1$ to $N$, $\ell = 1$ to $d$ **do**
  8:   $\mathbf{v}'_{i,\ell} \leftarrow \mathbf{J}_f(\mathbf{e}_i)\,\mathbf{v}_{i,\ell}$ {Jacobian from $f$, geometry from $f_{\mathrm{ref}}$}
  9: **end for**
 10: **Build coherence matrix:** $G_{ii} = 1$; for $j \in \mathcal{N}_i$: $G_{ij} = \frac{1}{d} \sum_\ell |\cos(\mathbf{v}'_{i,\ell}, \mathbf{v}'_{j,\ell})|$
 11: **Spectral analysis:** $p_i = |\lambda_i| / \sum_j |\lambda_j|, \quad S = -\sum_i p_i \log_2 p_i$
 12: **return** $\mathrm{GCS} = N - 2^S$

---

## 4 Experiments

To validate our geometric coherence framework and investigate the learning dynamics of Transformers, we conduct a series of controlled experiments on an algorithmic task known to exhibit grokking. This section details our experimental setup, including the task, model architecture, training protocol, and the specific configuration for our geometric coherence analysis.

### 4.1 Experimental Setup

The setup follows established protocols for mechanistic interpretability studies (Nanda et al., 2023; Liu et al., 2023a).

**Core Setup.** We focus on modular addition $c \equiv (a + b) \pmod{p}$ with prime modulus $p = 113$. The dataset consists of all $p^2 = 12{,}769$ input pairs, split into 30% training (3,830 pairs) and 70%

testing (8,939 pairs). Input sequences are $[\text{token}_a, \text{token}_b, \text{token}_{\text{equals}}]$, where the equals token is 113. We use a single-layer decoder-only Transformer with embedding dimension $d_{\text{model}} = 128$, multi-head attention ($n_{\text{heads}} = 4$, $d_{\text{head}} = 32$), and MLP with hidden dimension $d_{\text{mlp}} = 512$. The MLP consists of input weights $W_{\text{in}} \in \mathbb{R}^{d_{\text{mlp}} \times d_{\text{model}}}$, output weights $W_{\text{out}} \in \mathbb{R}^{d_{\text{model}} \times d_{\text{mlp}}}$, with unembedding matrix $W_U \in \mathbb{R}^{d_{\text{vocab}} \times d_{\text{model}}}$ where $d_{\text{vocab}} = 114$. We analyze networks with ReLU, GeLU, and SiLU activations, demonstrating that our findings hold across different nonlinearities (Section 4.3). No layer normalization or embedding tying is used. Training employs full-batch gradient descent for 20,000 steps using AdamW optimizer: learning rate $1 \times 10^{-3}$, weight decay $\lambda = 1.0$, betas $\beta = (0.9, 0.98)$. All experiments use fixed random seeds. While our primary analysis focuses on single-layer Transformers, we extend to 2-layer and 3-layer architectures in Section 4.4 and Appendix C.

**Geometric Analysis Configuration.** We employ the **Fixed Final Geometry** protocol (Algorithm 1): the geometric structure ($k$-NN graph, tangent bases, embeddings) is built once from the final model $f_{\text{ref}}$, while only the Jacobian $\mathbf{J}_f$ varies across checkpoints. This isolates the evolution of the learned transformation, measuring whether each checkpoint's Jacobian aligns tangent vectors consistently with the final model's manifold. We use $N = 200$ samples, $k = 15$ neighbors, $d = 8$ dimensions; robustness is confirmed in Appendix B. GCS is computed every 200 steps.

We analyze three Computational Flows at the final token position: **Attention Flow** (block input $\to$ attention output), **MLP Flow** (post-attention residual $\to$ MLP output), and **Overall Flow** (block input $\to$ block output).

## 4.2 A THREE-STAGE GEOMETRIC DYNAMIC IN GROKKING

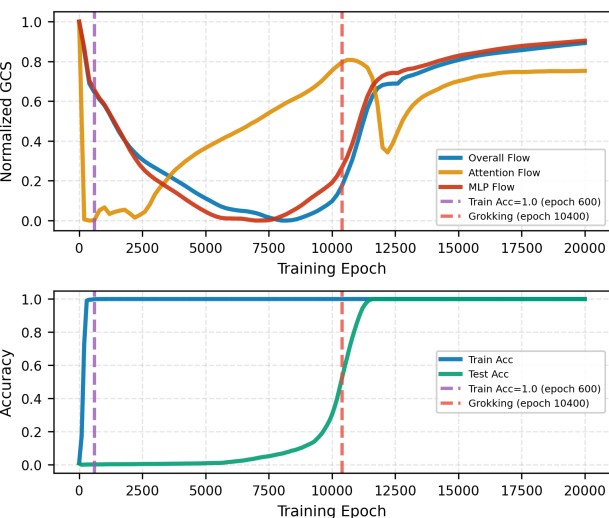

Figure 2: **The three-stage geometric evolution during grokking.** The top panel shows the test loss (red, log scale) and test accuracy (green), marking a sharp generalization transition around step 10,400. The bottom panel displays the corresponding evolution of the normalized Geometric Coherence Score (GCS) for three key computational flows.

Our central discovery, illustrated in Figure 2, is that the emergence of generalization in the Transformer is orchestrated by a remarkable, non-monotonic, **three-stage geometric learning dynamic** which we term **"construct-then-compress"** algorithm. This dynamic is characterized not by a simple sequence, but by a sophisticated, overlapping interplay between the Attention and MLP modules.

**Phase I: Memorization with Coherence Collapse (Steps 0–600).** Training begins with all flows exhibiting high GCS due to spurious uniformity in transformations. As the network memorizes the training data, all three flows descend concurrently, with the Attention flow declining most rapidly and reaching its minimum first, while the MLP and Overall flows continue to decrease. By the end of

this phase, the network achieves perfect training accuracy, indicating complete memorization. However, all pathways have abandoned their initial trivial coherence, setting the stage for algorithmic discovery.

**Phase II: Asynchronous Construction then Compression (Steps 600–10,400).** This extended phase reveals asynchronous coordination among the pathways. The Attention flow, having reached its minimum first, initiates compression earliest, rising steadily as it constructs structured attention patterns. The MLP and Overall flows reach their minima later and then begin their ascent. Both pathways compress simultaneously but with a temporal offset: Attention leads, and MLP follows. This phase culminates as the Attention flow approaches its peak coherence while MLP compression accelerates—marking the grokking transition, where test accuracy exhibits its steepest rise, though not yet reaching saturation.

**Phase III: Post-Grokking Refinement (Steps 10,400+).** Following the grokking transition, all pathways undergo continued refinement that brings test accuracy to full saturation. Most notably, the Attention flow exhibits a characteristic *double descent*—a secondary drop in coherence following its Phase II peak—suggesting algorithmic fine-tuning as attention patterns are adjusted to better align with the discovered solution. The MLP and Overall flows stabilize at high coherence with minor adjustments. Through this refinement phase, test accuracy completes its rise to near 100%, and the network converges to its final, geometrically coherent algorithm.

### 4.3 ROBUSTNESS OF THE THREE-STAGE DYNAMIC

Having identified the three-stage dynamic in a single-layer Transformer, we now confirm its robustness across diverse experimental conditions. The central question is whether this geometric choreography—Phase I coherence collapse, Phase II asynchronous construction-then-compression with Attention leading, and Phase III post-grokking refinement—represents a fundamental property of Transformer learning.

**Consistency Across Activation Functions.** We repeated our analysis using three activation functions (ReLU, GeLU, SiLU) on the modular addition task. Figure 3 reveals remarkable qualitative consistency: all three exhibit the same three-stage structure with identical temporal ordering (Attention leads construction and compression, MLP follows), despite notable quantitative differences in learning speed and final GCS magnitude. ReLU networks grok fastest; SiLU networks are slowest. This invariance demonstrates that the three-stage dynamic emerges from the Transformer's architectural inductive biases rather than particular nonlinear choices.

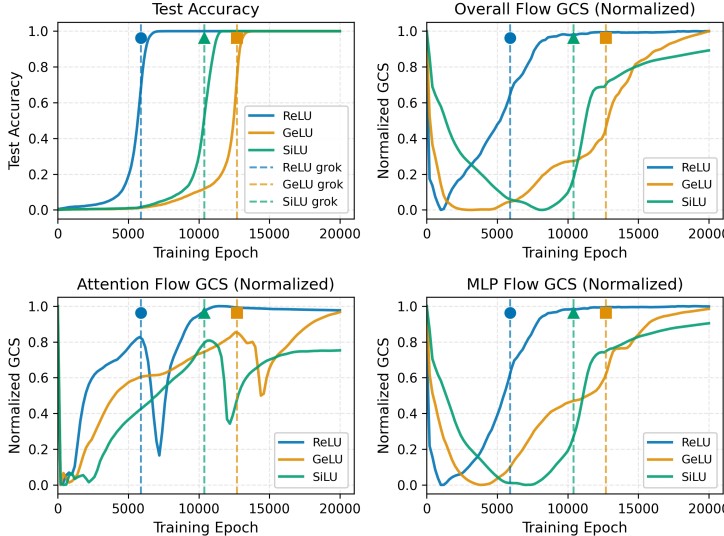

Figure 3: **Impact of Activation Functions on the Three-Stage Dynamic.** The construct-then-compress choreography (Phase I–II–III) persists across ReLU, GeLU, and SiLU despite timing variations, where Attention leads MLP in geometric reorganization.

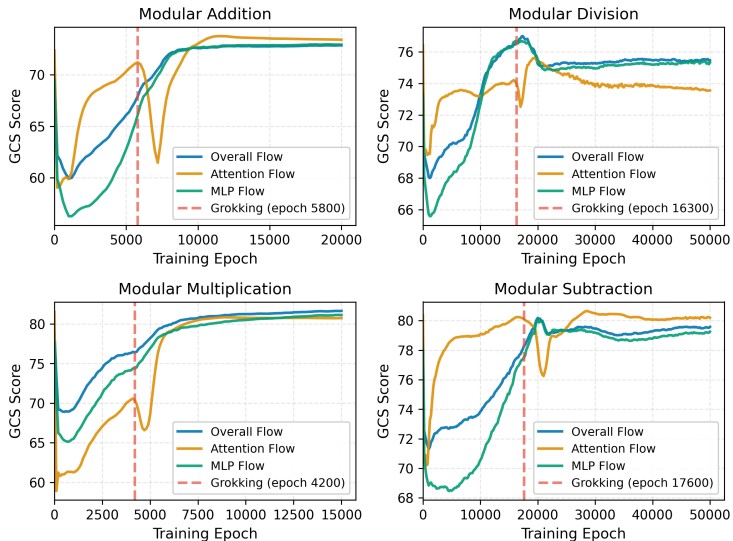

Figure 4: **Impact of Modular Tasks on the Three-Stage Dynamic.** While symmetric tasks (add, mul) show monotonic compression in Phase III, asymmetric tasks (div, sub) exhibit MLP double descent, reflecting higher algorithmic complexity.

**Consistency Across Modular Operations.** Extending our analysis to all four fundamental modular operations (addition, subtraction, multiplication, division), we find that the three-stage dynamic is fully preserved across all tasks (Figure 4). Every operation exhibits the same temporal choreography: Phase I coherence collapse during memorization, Phase II asynchronous construction-then-compression with Attention leading, and Phase III post-grokking refinement.

The only task-dependent variation occurs in the *refinement pattern* of Phase III. Symmetric operations (addition, multiplication) show continued MLP compression—a monotonic rise toward stable high coherence. In contrast, asymmetric operations (division, subtraction) exhibit MLP *double descent* alongside the Attention double descent, suggesting that these algorithmically more complex tasks require additional geometric fine-tuning across *both* pathways before converging to their final solution. This variation enriches rather than contradicts our framework: the three-stage structure accommodates task-specific refinement dynamics while maintaining its core temporal choreography.

## 4.4 HIERARCHICAL ORGANIZATION IN MULTI-LAYER TRANSFORMERS

We extend our analysis to 2-layer and 3-layer Transformers on modular addition ($p = 113$), with detailed layer-wise GCS trajectories provided in Appendix C. An interesting pattern emerges: as depth increases, the final layer achieves *lower* geometric coherence while early layers maintain higher stability (Table 1).

Table 1: Final geometric coherence (GCS at convergence) on modular addition ($p = 113$). Deeper networks show progressively lower final-layer GCS, particularly in attention paths, suggesting more specialized geometric transformations in later layers.

| Architecture | Layer | Overall | Attention | MLP |
|---|---|---|---|---|
| 1-Layer | — | 74.9 | 74.9 | 74.8 |
| 2-Layer | Layer 1 | 87.0 | 85.9 | 87.0 |
| | Layer 2 | 66.8 | 64.4 | 68.8 |
| 3-Layer | Layer 1 | 68.8 | 68.7 | 68.7 |
| | Layer 2 | 71.4 | 64.3 | 71.4 |
| | Layer 3 | 61.0 | 58.6 | 62.5 |

**Depth-Dependent Geometric Specialization.** The progressive decrease in final-layer attention GCS—from 74.9 (1-layer) to 64.4 (2-layer) to 58.6 (3-layer)—reveals a consistent pattern: deeper networks employ increasingly *specialized* geometric transformations in their final layers, while early layers maintain higher coherence that provides stable representational foundations. This stratification suggests that multi-layer networks decompose the modular arithmetic task hierarchically, with early layers establishing structured features and final layers performing more input-specific computations. The potential connection between this geometric specialization and depth-dependent algorithmic complexity (e.g., frequency utilization) remains an open question discussed in Section 5.

**Path-Specific Dynamics in Hierarchical Processing.** Table 2 reveals a nuanced pattern of geometric restructuring across layers and paths. While the overall and MLP paths in early layers remain nearly static (ranges 0.2–1.3), attention dynamics vary by depth: in 2-layer networks, the early layer shows substantial attention restructuring (range 13.0), while in 3-layer networks, attention dynamics distribute across middle layers (Layer 2 range 7.4) with the earliest layer remaining nearly static. In contrast, final layers show substantial restructuring across all paths. Notably, unlike the single three-phase trajectory observed in 1-layer networks, multi-layer final layers exhibit *iterative construct-compress cycles*—alternating phases of coherence increase and decrease—suggesting that hierarchical processing involves repeated refinement rather than a single pass. This path-specific division of labor, where early layers maintain stable MLP transformations while final layers undergo iterative geometric reorganization, reveals depth-dependent dynamics distinct from single-layer behavior.

Table 2: Geometric restructuring magnitude (GCS range) during training. Early layers show path-specific stability (overall and MLP nearly static); attention dynamics vary by depth—substantial in 2-layer (13.0), distributed to middle layers in 3-layer (7.4). Final layers show substantial restructuring: 1-layer exhibits clear three-stage dynamics, while multi-layer final layers show iterative construct-compress cycles.

| Architecture | Layer | Overall | Attention | MLP | Pattern |
|---|---|---|---|---|---|
| 1-Layer | — | 4.6 | 17.9 | 7.2 | Three-stage |
| 2-Layer | Layer 1 | 0.2 | 13.0 | 0.2 | Stable (Attn dynamic) |
| | Layer 2 | 10.0 | 12.9 | 14.5 | Iterative cycles |
| 3-Layer | Layer 1 | 1.3 | 1.0 | 1.2 | Nearly static |
| | Layer 2 | 0.8 | 7.4 | 0.8 | Stable (Attn dynamic) |
| | Layer 3 | 9.7 | 6.2 | 14.5 | Iterative cycles |

## 5 DISCUSSION

**Universality of Geometric Dynamics.** By extending our analysis across diverse tasks and architectures, we establish that the construct-then-compress principle is robust across activation functions and modular operations. In single-layer networks, this manifests as a clear three-phase evolution; in multi-layer networks, final layers exhibit iterative construct-compress cycles, suggesting that hierarchical processing involves repeated refinement. This supports the "lazy-to-rich" framework (Chou et al., 2025; Kumar et al., 2024), but adds geometric precision: the "richness" is specifically the construction of coherent transformations, with depth introducing iterative refinement dynamics.

**Geometric Grounding of Competing Circuits.** GCS provides a physical basis for the abstract competing circuits hypothesis (Merrill et al., 2023). We interpret the memorization circuit as geometrically incoherent (disjoint Jacobians) and the generalization circuit as coherent (aligned Jacobians). The steady rise of GCS during the accuracy plateau (Phase II) acts as an early warning system, visualizing the silent growth of the generalization circuit before it dominates behavior. This connects time-wise grokking to model-wise double descent (Davies et al., 2022), identifying their shared geometric origin.

**GCS vs. Dimensionality Metrics.** A key methodological contribution is the distinction between geometric coherence and representational dimensionality. During Phase II, we observe "iso-dimensional" organization: the Participation Ratio remains flat (indicating stable global dimension-

ality) while GCS rises steadily (indicating increasing algorithmic coherence). This demonstrates that networks can reorganize their internal mechanisms to be more coherent *without* changing their representational capacity—a phenomenon invisible to purely dimensional metrics (Appendix F).

**Scalability and Hierarchical Structure.** Our results on deeper networks reveal hierarchical organization: early layers establish path-specific stability (stable MLP transformations with depth-varying attention dynamics), while final layers undergo iterative construct-compress cycles distinct from the single-pass three-phase pattern of shallow networks. Understanding why depth induces this iterative refinement remains an open question. GCS is computationally efficient—relying on local $k$-NN and SVD operations rather than global Hessian computation, scaling linearly with sample size $N$ and layer count $L$ ($O(NL)$)—making it practical for analyzing larger models.

**Limitations and Future Work.** Methodologically, our variance-based alignment relies on singular vector ordering stability. While theoretically ambiguous in isotropic distributions, this is mitigated by the strong anisotropy in neural representations (Ethayarajh, 2019). Future work could explore permutation-invariant metrics (e.g., Grassmannian distance) for degenerate cases. Empirically, we focused on standard Transformers with smooth activations. Investigating pure MLPs or quadratic activations (Gromov, 2023), and extending this geometric lens to semantic coherence in Large Language Models, remain promising frontiers. Our layer-wise GCS measurements also complement prior work on depth-dependent algorithm learning (Morwani et al., 2024; McCracken et al., 2025)—whether lower GCS *enables* or merely *correlates with* reduced frequency requirements remains an open question connecting geometric and spectral perspectives.

**Conclusion.** We introduced the Geometric Coherence Score (GCS), a principled metric grounded in differential geometry that quantifies the consistency of a network's learned transformations. Applying GCS to grokking, we discovered a construct-then-compress dynamic that orchestrates the transition from memorization to generalization. In single-layer networks, this manifests as a clear three-stage evolution; in multi-layer networks, final layers exhibit iterative construct-compress cycles, revealing depth-dependent dynamics that warrant further investigation. The construct-then-compress principle is robust across activation functions and modular tasks, providing direct geometric evidence for how Transformers discover algorithmic solutions. By revealing that both Attention and MLP pathways actively participate in geometric reorganization, our work establishes GCS as a diagnostic tool for interpreting emergent phenomena in neural networks.

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

## A  THEORETICAL DERIVATION OF GCS BOUNDS

To validate the physical interpretation of the Geometric Coherence Score (GCS), we derive its behavior in two theoretical limit cases: Total Geometric Incoherence (representing pure memorization) and Perfect Geometric Coherence (representing ideal algorithmic unification). This derivation demonstrates that GCS functions as a rigorous measure of complexity reduction.

### A.1  CASE 1: TOTAL GEOMETRIC INCOHERENCE (THE MEMORIZATION LIMIT)

Consider a network in a state of pure memorization, where each data point is processed independently. In this regime, the local geometric transformation at any point $\mathbf{x}_i$ is uncorrelated with the transformation at its neighbor $\mathbf{x}_j$. Consequently, the tangent vectors become orthogonal or randomly oriented in the high-dimensional space.

**Mathematical Formulation:**  The pairwise geometric similarity $G_{ij}$ approaches zero for all distinct pairs, while self-similarity remains unity:

$$G_{ij} \approx \delta_{ij} = \begin{cases} 1 & \text{if } i = j \\ 0 & \text{if } i \neq j \end{cases} \tag{5}$$

Thus, the coherence matrix $\mathbf{G}$ approximates the identity matrix $\mathbf{I}_N \in \mathbb{R}^{N \times N}$.

**Spectral Analysis:** The eigenvalues of the identity matrix satisfy $\det(\mathbf{I}_N - \lambda\mathbf{I}_N) = (1-\lambda)^N = 0$. Hence, the spectrum is perfectly degenerate:

$$\lambda_1 = \lambda_2 = \cdots = \lambda_N = 1 \tag{6}$$

We normalize this spectrum to obtain the probability distribution $p$:

$$p_i = \frac{\lambda_i}{\sum_{j=1}^{N} \lambda_j} = \frac{1}{N}, \quad \forall i \in \{1, \ldots, N\} \tag{7}$$

This yields a uniform distribution over the geometric modes.

**GCS Computation:** The Von Neumann entropy $S_{\mathrm{NL}}$ is maximized for the uniform distribution:

$$S_{\mathrm{NL}} = -\sum_{i=1}^{N} p_i \log_2 p_i = -\sum_{i=1}^{N} \frac{1}{N} \log_2\left(\frac{1}{N}\right) = \log_2 N \tag{8}$$

The Geodesic Mode Number (GMN) and GCS are derived as:

$$\mathrm{GMN} = 2^{S_{\mathrm{NL}}} = 2^{\log_2 N} = N \tag{9}$$
$$\mathrm{GCS} = N - \mathrm{GMN} = N - N = 0 \tag{10}$$

**Conclusion:** In the limit of total incoherence, the network exhibits $N$ independent geometric degrees of freedom, resulting in a GCS of exactly 0.

## A.2 CASE 2: PERFECT GEOMETRIC COHERENCE (THE ALGORITHMIC LIMIT)

Consider a network that has discovered a unified, generalizable algorithm (e.g., a consistent rotation across a manifold). In this ideal limit, the network applies an identical geometric transformation to all points, resulting in perfect alignment between all local tangent spaces.

**Mathematical Formulation:** In the theoretical limit where all points exhibit identical transformations (relaxing the k-NN constraint for analytical purposes), the geometric similarity between any pair of points is maximal. The coherence matrix $\mathbf{G}$ approaches the all-ones matrix $\mathbf{J}_N$ (where $G_{ij} = 1, \forall i, j$).

**Spectral Analysis:** The all-ones matrix $\mathbf{J}_N$ has rank 1. To find its eigenvalues, note that $\mathbf{J}_N \mathbf{v} = N\mathbf{v}$ when $\mathbf{v} = [1, 1, \ldots, 1]^\top$, while any vector orthogonal to $\mathbf{v}$ is mapped to zero. Thus:

$$\lambda_1 = N, \quad \lambda_2 = \cdots = \lambda_N = 0 \tag{11}$$

The normalized probability distribution $p$ becomes a Kronecker delta distribution (pure state):

$$p_1 = \frac{N}{N} = 1, \quad p_i = \frac{0}{N} = 0 \text{ for } i > 1 \tag{12}$$

**GCS Computation:** The entropy of this pure state vanishes:

$$S_{\mathrm{NL}} = -1 \cdot \log_2(1) - \sum_{i=2}^{N} 0 \cdot \log_2(0) = 0 \tag{13}$$

where we use the convention $0 \log_2(0) = 0$. The GMN and GCS are derived as:

$$\text{GMN} = 2^0 = 1 \tag{14}$$

$$\text{GCS} = N - 1 \tag{15}$$

**Conclusion:** In the limit of perfect coherence, the network's geometric behavior collapses into a single effective mode (GMN $= 1$), resulting in a maximal GCS of $N - 1$.

### A.3    INTERPRETATION

These derivations confirm that GCS $= N - $ GMN serves as a linear measure of complexity reduction. It quantifies the number of redundant geometric degrees of freedom the network has successfully eliminated, ranging from 0 (chaos/memorization) to $N - 1$ (order/algorithmic discovery). The Von Neumann entropy, borrowed from quantum information theory, naturally captures the effective dimensionality of the geometric transformation space, making GCS a principled measure of algorithmic compression.

## B    HYPERPARAMETER ROBUSTNESS

A critical aspect of our geometric analysis is ensuring that the observed learning dynamics are not artifacts of specific hyperparameter choices. We evaluate the sensitivity of the Geometric Coherence Score (GCS) to two key parameters: the intrinsic dimension $d$ of the local tangent spaces, and the number of evaluation samples $N$. Our robustness analysis demonstrate that the core geometric phenomena—specifically the three-stage dynamic and the grokking transition—are robust across wide parameter ranges.

### B.1    ROBUSTNESS TO INTRINSIC DIMENSION $d$

The intrinsic dimension $d$ determines the rank of the local linear approximation used to probe the network's geometry. To understand its impact, we conducted experiments with $d \in \{2, 4, 6, 8, 10, 12, 14\}$, keeping the neighborhood size fixed at $k = 15$ and sample size at $N = 200$.

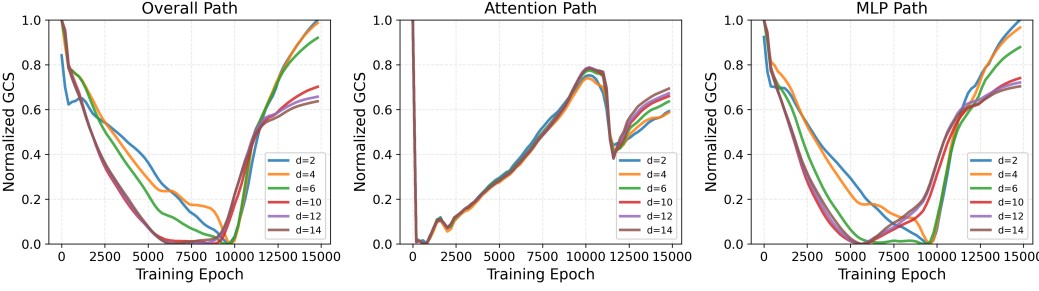

Figure 5: **Intrinsic Dimension Robustness. (Left)** Normalized GCS curves for the Attention Flow across varying dimensions $d$. The characteristic U-shape dynamic is preserved universally. **(Right)** Pairwise correlation matrix of GCS trajectories between different $d$ values. The high correlation ($r > 0.90$) confirms that different dimensions capture the same underlying geometric evolution.

As shown in Figure 5, our analysis reveals strong consistency across dimensions:

**Shape Consistency:** All tested dimensions $d$ produce highly congruent GCS trajectories. The characteristic "U-shape" curve—marking the transition from memorization to construction and finally compression—is clearly visible in all cases. **High Correlation:** We computed the Pearson correlation coefficient between the GCS trajectories of different $d$ values. The cross-dimension correlations consistently exceed $0.90$, with an average correlation of $0.97$ relative to our chosen baseline of $d = 8$. This confirms that low-dimensional probes ($d = 2$) and higher-dimensional probes ($d = 14$) are measuring the same fundamental geometric process. **Selection of** $d = 8$**:** While lower dimensions ($d = 2$) exhibit a higher dynamic range (sensitivity), they risk underspecifying the geometric complexity of the 128-dimensional representation space. Conversely, excessively high dimensions may

introduce noise. We selected $d = 8$ for the main experiments as a **conservative middle ground** that balances signal sensitivity with sufficient representational capacity to capture complex local structures.

## B.2 Robustness to Sample Size $N$

We further evaluated the stability of our metric with respect to the sample size $N$ used for the geometric reference frame. We tested $N \in \{100, 200, 400, 750, 1850\}$ with fixed $d = 8$ and $k = 15$.

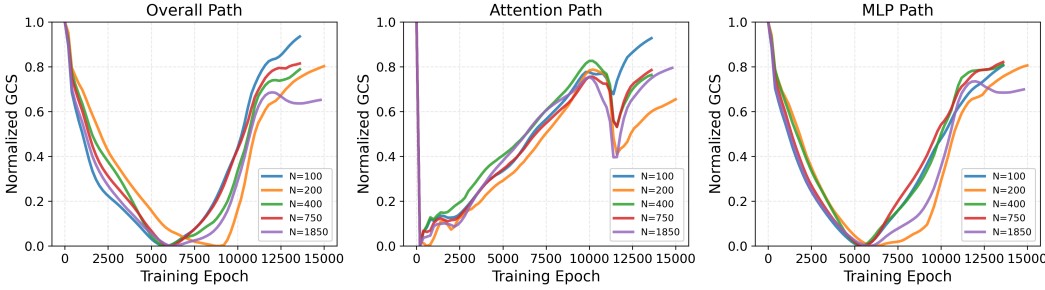

Figure 6: **Sample Size Robustness.** Normalized GCS dynamics for varying sample sizes $N$. The core three-stage pattern and the timing of the grokking transition are robustly detected even at $N = 100$. Larger sample sizes provide smoother measurements but do not alter the qualitative findings.

Figure 6 demonstrates that the detected learning dynamics are not sensitive to sample size:

**Robust Detection:** The three-stage dynamic and the precise timing of the grokking transition are accurately captured even with as few as $N = 100$ samples. **Convergence:** As $N$ increases, the GCS trajectories become smoother, but the qualitative behavior and relative ordering of the computational flows (Attention vs. MLP) remain unchanged. **Efficiency:** Based on these results, we employed $N = 200$ for our main experiments. This choice provides a reliable, low-variance estimation of geometric coherence while maintaining high computational efficiency, allowing for dense monitoring of the training process.

## B.3 Summary

These robustness studies confirm that the "Construct-then-Compress" mechanism is a robust feature of the network's learning dynamics, invariant to specific hyperparameter choices. The high cross-parameter correlations indicate that GCS captures an intrinsic geometric property of the learning process rather than an artifact of the measurement setup.

## C Multi-Layer Transformer Dynamics

This appendix provides detailed visualizations of layer-wise GCS trajectories for 2-layer and 3-layer Transformers, complementing the quantitative analysis in Section 4.4.

## C.1 2-Layer Transformer Dynamics

Figure 7 presents a comprehensive view of 2-layer Transformer training on modular addition ($p = 113$). The top-right panel shows the learning dynamics, with grokking occurring around epoch 12,000 (marked by red dashed line). The top-left and top-middle panels compare GCS trajectories for Layer 1 and Layer 2 across all three computational flows. Layer 1 exhibits path-specific stability: the overall and MLP flows maintain nearly constant GCS throughout training (range $\sim 0.2$), serving as stable feature transformations, while the attention flow shows substantial dynamics (range $\sim 13.0$), suggesting adaptive routing mechanisms even in early layers. Layer 2 exhibits iterative construct-compress cycles across all flows—alternating phases of coherence increase and decrease—distinct from the single three-phase trajectory observed in 1-layer networks. This suggests that hierarchical processing involves repeated refinement rather than a single pass. The bottom panels show

flow-specific layer comparisons, clearly demonstrating that comprehensive geometric reorganization concentrates in the final layer.

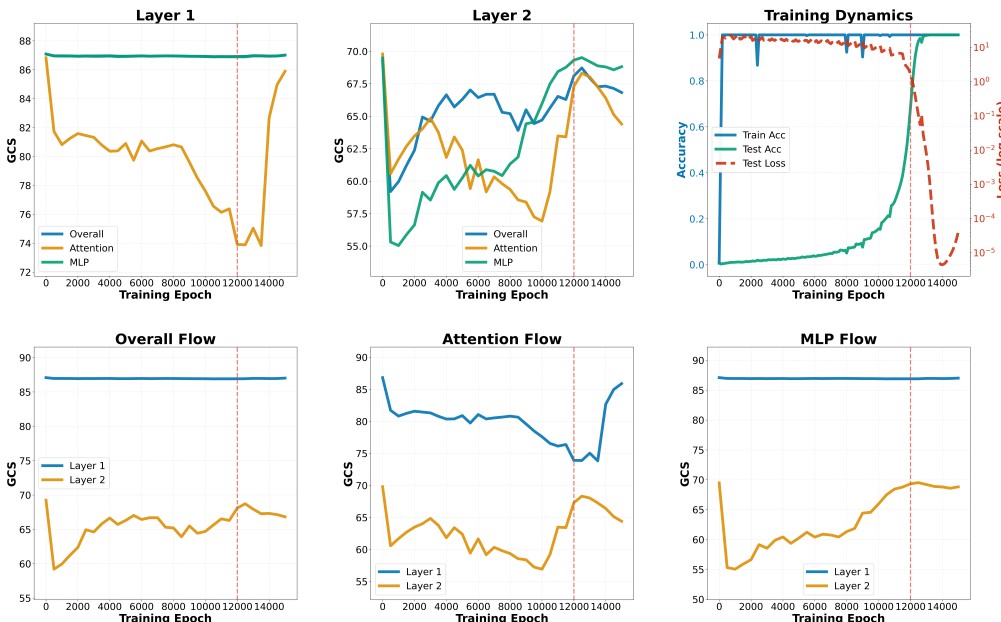

Figure 7: Comprehensive dynamics of 2-layer GeLU Transformer on modular addition. **Top-right:** Learning curves showing grokking transition. **Top-left and Top-middle:** Layer-wise GCS trajectories showing Layer 1 path-specific stability (overall/MLP stable at ∼87%, attention substantially dynamic) versus Layer 2 iterative construct-compress cycles across all paths. **Bottom:** Flow-specific layer comparisons (Overall, Attention, MLP) highlighting the concentration of geometric reorganization in Layer 2. Red dashed line marks grokking point (epoch 12,000).

## C.2 3-LAYER TRANSFORMER: PROGRESSIVE STRATIFICATION

Figure 8 reveals the progressive stratification pattern in 3-layer networks. The top row displays per-layer GCS trajectories, showing increasingly complex dynamics with depth. Layer 1 maintains high stability across all paths (overall, MLP, and attention all with range ∼1.0–1.3), remaining nearly static throughout training. Layer 2 exhibits path-specific patterns: overall and MLP paths remain stable (range ∼0.8) while the attention path shows moderate dynamics (range ∼7.4), indicating evolving routing strategies in intermediate processing. Layer 3 exhibits iterative construct-compress cycles across all paths (range ∼6–15)—alternating phases of coherence increase and decrease—consistent with the pattern observed in 2-layer networks but distinct from the single three-phase trajectory of 1-layer networks. The bottom row presents flow-specific layer comparisons, confirming that early layers provide stable feature bases while attention adaptation progressively shifts from early layers (2-layer) to middle layers (3-layer), with final layers undergoing iterative geometric reorganization.

# D ON SVD-ORDERED SINGULAR VECTOR CORRESPONDENCE

This appendix clarifies the design choice in Equation (3): using row-wise correspondence between singular vectors $\mathbf{v}_{i,k}$ and $\mathbf{v}_{j,k}$. We explain that (1) this order-sensitivity is intentional and geometrically meaningful, and (2) the potential instability from degenerate singular values does not occur in practice.

## D.1 THE ORDER-SENSITIVITY IS INTENTIONAL

Our metric deliberately uses SVD-ordered correspondence rather than order-invariant subspace measures (e.g., principal angles). This is a design choice, not an oversight:

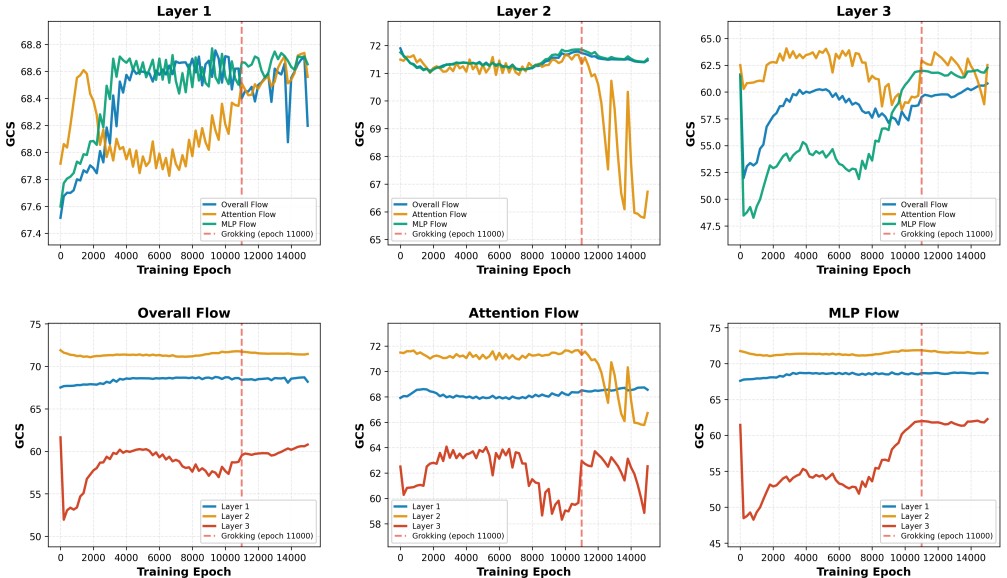

Figure 8: Progressive stratification in 3-layer GeLU Transformer on modular addition. **Top row:** Per-layer GCS showing increasing dynamical complexity from Layer 1 (nearly static across all paths) through Layer 2 (path-specific: attention moderately adaptive, overall/MLP stable) to Layer 3 (iterative construct-compress cycles). **Bottom row:** Flow-specific layer comparisons revealing the hierarchical organization of geometric restructuring, with iterative reorganization concentrated in Layer 3. Red dashed line marks grokking point (epoch 11,000).

**We measure data manifold coherence, not abstract subspace overlap.**

SVD orders singular vectors by variance magnitude—the $k$-th vector represents the "$k$-th most important direction" at each point. By comparing $\mathbf{v}_{i,k}$ with $\mathbf{v}_{j,k}$, we ask: *do nearby points on the manifold share similar principal geometric structure?*

This captures richer information than subspace overlap alone:

- Two points may span similar subspaces but with *different* principal directions (low $G_{ij}$)
- Two points may have *aligned* principal hierarchies indicating coherent local geometry (high $G_{ij}$)

The distinction matters for detecting whether a network has learned a *consistent geometric algorithm* versus merely preserving some abstract subspace structure.

### D.2    EMPIRICAL VALIDATION: NEURAL REPRESENTATIONS ARE ANISOTROPIC

If singular values were nearly equal (isotropic geometry), SVD ordering would be numerically unstable. However, neural network representations are known to be highly anisotropic—occupying narrow cones rather than uniform spheres in representation space (Ethayarajh, 2019; Martin & Mahoney, 2021). We verify this property holds in our experiments.

We analyzed 200 sampled points per model across three activation functions (ReLU, SiLU, GeLU), all trained to 100% test accuracy on modular addition.

Table 3: Singular value separation confirms strong anisotropy across all models (N=200 per model).

| Metric | ReLU | SiLU | GeLU |
|---|---|---|---|
| Condition number $\sigma_1/\sigma_d$ | $5.0 \pm 0.9$ | $7.6 \pm 1.7$ | $5.1 \pm 0.9$ |
| Isotropic points (cond. $< 2$) | 0 | 0 | 0 |

**Key findings:**

- All 600 points have condition number $> 3$, with means of 5–7
- A condition number of 5 means $\sigma_1$ is $5\times$ larger than $\sigma_d$—unambiguously anisotropic
- Zero points exhibit near-isotropic geometry where ordering would be ambiguous

### D.3 CONCLUSION

The order-sensitive design in Equation (3) is rational: it measures manifold coherence rather than abstract subspace similarity. The theoretical concern about degenerate singular values causing unstable ordering does not apply to neural network representations, which are strongly anisotropic (condition numbers $5$–$7\times$ in our experiments, with zero isotropic points).

## E A CONTRASTING CASE: THE GEOMETRIC SIGNATURE OF OVERFITTING

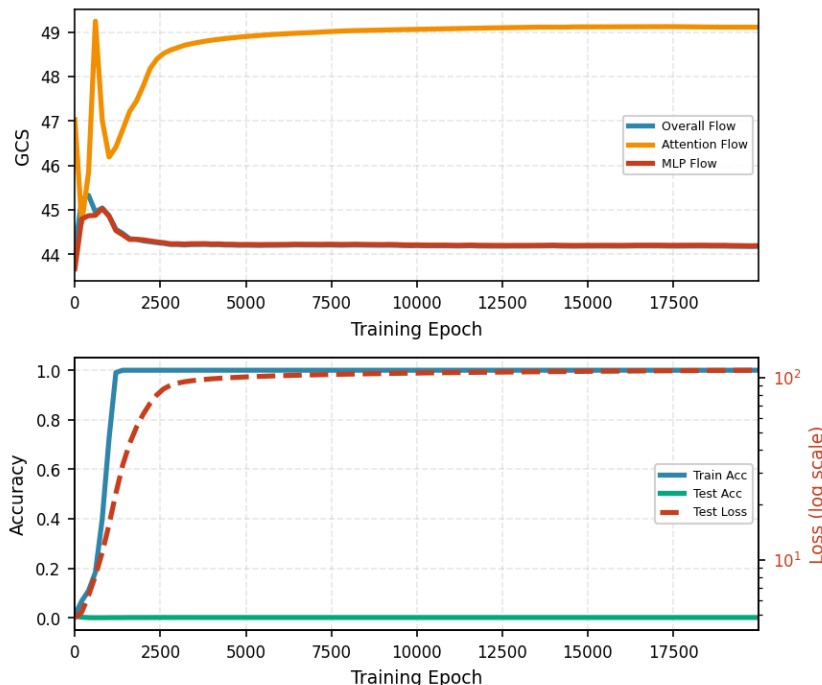

Figure 9: Overfitting exhibits fundamentally different geometric dynamics. The absence of Phase II complexity construction prevents the "construct-then-compress" mechanism, resulting in persistent memorization without generalization.

To demonstrate that the "Construct-then-Compress" dynamic is a specific signature of algorithmic generalization rather than a generic training artifact, we analyze a counter-factual scenario: a network that successfully memorizes the training data but fails to generalize (Overfitting). Figure 9 illustrates the geometric evolution of a 1-Layer Transformer (SeLU activation) trained on a small subset of data, which achieves 100

**Premature Compression in Attention.** In a successful grokking trajectory, the Attention mechanism typically undergoes a coherence collapse" (Phase I & early Phase II) to break symmetry and construct rich features. However, in the overfitting regime, we observe a distinct deviation:

- Instead of entering the **Construction Phase**, the Attention Flow GCS spikes and remains persistently high (top panel, orange line).
- This indicates that the Attention mechanism has bypassed the necessary step of geometric restructuring. It effectively settles into a lazy" solution (e.g., relying solely on positional

embeddings or identity mappings) that maintains high geometric triviality but fails to extract task-relevant topology.

**Consequential Stagnation in MLP.** Because the Attention head performs Premature Compression" without extracting meaningful algorithmic features, the downstream MLP is forced to memorize the residuals.

- The MLP Flow (red line) never exhibits the characteristic U-shaped Construct-then-Compress" trajectory.
- Instead, it remains flat at a low coherence level throughout training. This confirms that without the upstream construction of structured representations by Attention, the MLP cannot perform the subsequent compression required for generalization.

**Diagnostic Value.** Crucially, this failure mode is detectable as early as Phase II. While the training accuracy (blue line, bottom panel) rises perfectly, the divergence in GCS dynamics—specifically the **absence of the construction dip in Attention** and the **absence of the compression rise in MLP**—signals that the model is on a trajectory toward overfitting. This validates GCS as a falsifiable metric: the monotonic "Construct-then-Compress" dynamic is not inevitable, but suggests the specific causal mechanism of algorithmic discovery.

# F COMPARATIVE ANALYSIS: GEOMETRIC COHERENCE VS. PARTICIPATION RATIO

To ascertain whether the Geometric Coherence Score (GCS) provides novel mechanistic insights beyond existing measures of representational geometry, we conducted a side-by-side comparison with the Participation Ratio (PR)(Murphy et al., 2011). PR is a widely used metric for quantifying the effective dimensionality of neural representations.

## F.1 THEORETICAL DISTINCTION

While both metrics characterize the geometry of the network, they measure fundamentally orthogonal properties:

- **Participation Ratio (Static Representation Geometry):** PR is derived from the covariance matrix of the activations $\mathbf{X}$. It quantifies the *volume* or effective number of active dimensions utilized by the data distribution. It asks: *"How spread out is the data?"*

$$\text{PR}(\mathbf{C}) = \frac{(\text{Tr}(\mathbf{C}))^2}{\text{Tr}(\mathbf{C}^2)}, \quad \text{where } \mathbf{C} = \mathbb{E}\left[(\mathbf{x} - \boldsymbol{\mu})(\mathbf{x} - \boldsymbol{\mu})^\top\right] \quad (16)$$

- **Geometric Coherence Score (Dynamic Transformational Geometry):** GCS is derived from the Jacobian-Vector Products of the layer function $f$. It quantifies the *consistency* of the transformation applied to the data. It asks: *"Does the network process neighbors using the same algorithmic rule?"*

A network can have a high dimensional representation (High PR) that is processed chaotically (Low GCS), or a low dimensional representation (Low PR) processed coherently (High GCS).

## F.2 EMPIRICAL DIVERGENCE ACROSS TRAINING PHASES

We tracked both metrics throughout the training of a Transformer on modular addition. As shown in Figure 10, while both metrics reflect the global "expansion-then-compression" trend, GCS reveals critical mechanistic dynamics that PR misses or conflates. We observe three key divergences:

**1. Phase I: Distinguishing Chaos from Construction.** During early training, PR indicates a rapid expansion of dimensionality. However, dimensionality expansion is ambiguous: it can result from structured feature creation or random noise injection.

- **PR:** Monotonic increase (Expansion).

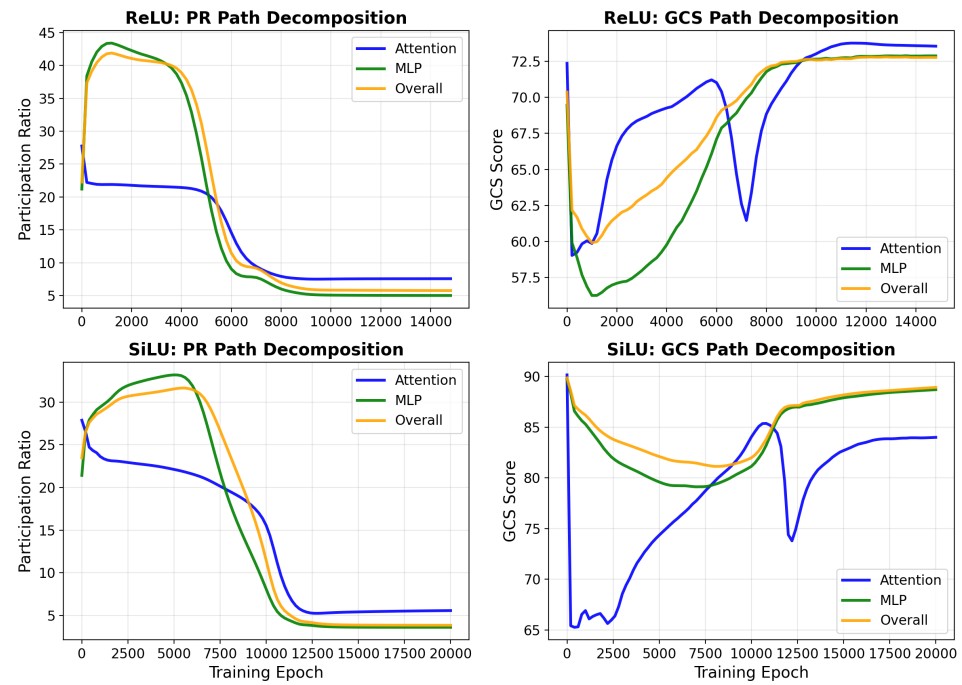

Figure 10: PR vs GCS across Activations(ReLU, SiLU)

  • **GCS:** Sharp decline (Coherence Collapse).

This divergence clarifies that the dimensionality increase corresponds to symmetry breaking—the network is actively sacrificing geometric uniformity to memorize disjoint data points. GCS resolves the ambiguity of "expansion" by detecting the loss of algorithmic structure.

**2. Phase II: Detecting "Iso-dimensional" Organization.**    This is the most significant divergence. During the "Silent Phase" where test accuracy is flat, we observe a period where the representation's global shape stabilizes, but the internal mechanism continues to evolve.

  • **PR:** Remains effectively flat/stable (indicating static global dimensionality).
  • **GCS (Attention Flow):** Rises steadily.

This reveals an "Iso-dimensional" organization phase: the network is actively refining the geometry of its attention mechanism—optimizing how information is routed—without changing the number of active dimensions. A purely dimensional metric misses this critical algorithmic alignment entirely.

**3. Phase III: Capturing Algorithmic Refinement.**    In the post-grokking phase, the network fine-tunes its solution.

  • **PR:** Flattens out, suggesting convergence.
  • **GCS:** Captures a characteristic "Double Descent" in the Attention flow (a secondary drop followed by recovery).

This signal correlates with the final ascent in test accuracy, indicating that the network continues to refine its geometric mechanism (e.g., pruning redundant modular artifacts) even after the effective dimensionality has stabilized.

F.3    CONCLUSION

These findings demonstrate that GCS is not a proxy for dimensionality. While PR measures the capacity of the representation space, GCS measures the coherence of the computational mecha-

nism. The "Iso-dimensional" evolution observed in Phase II confirms that measuring the Jacobian's geometric consistency provides a necessary, complementary lens for detecting the emergence of generalized algorithms.

## G    THE EVOLUTION OF ATTENTION PATTERNS

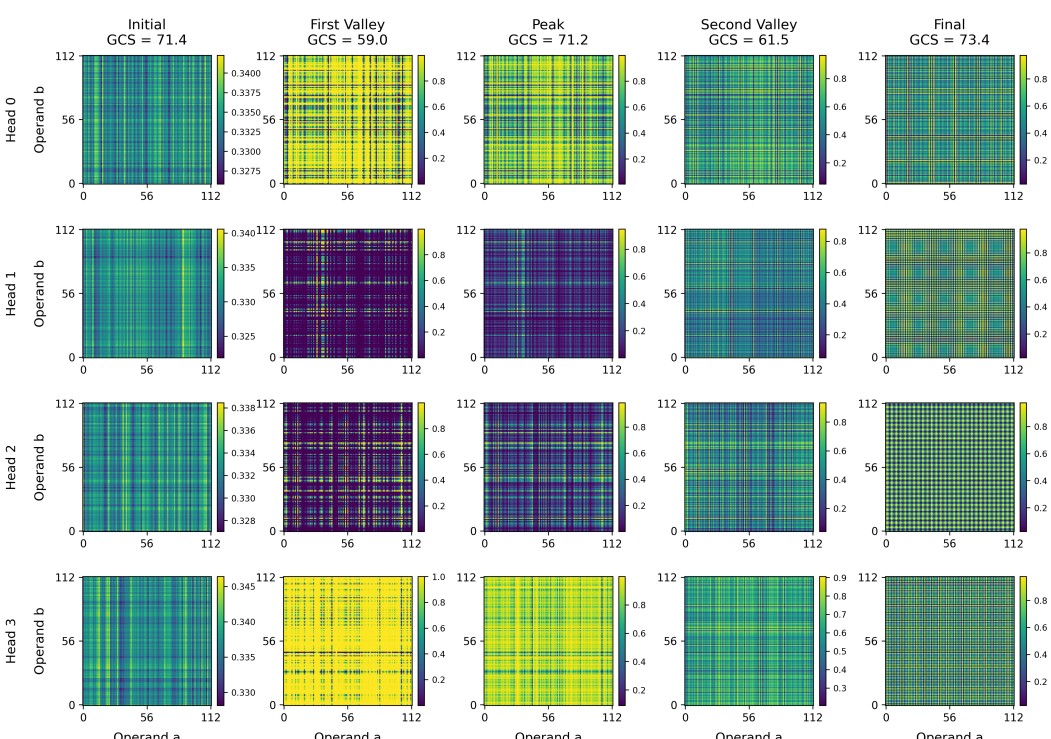

Figure 11: Mechanistic evolution of attention patterns across critical phases of the GCS trajectory. Heatmaps visualize the attention weights from the '=' token to the first operand 'a' across all input pairs $(a, b)$. Each column represents a key checkpoint defined by GCS extrema, and each row corresponds to one of the four attention heads. (a) First Valley (Memorization): GCS collapses ($\sim$59.0) as heads exhibit functional divergence; some heads (e.g., Head 1, 2) develop sparse, disjoint patterns to memorize specific outliers, disrupting geometric coherence. (b) Peak (Grokking): GCS recovers ($\sim$71.2) as all heads converge to a unified algorithmic strategy, characterized by clear diagonal (identity) and grid-like (modular) structures. (c) Final State: The patterns stabilize into a crystal-clear, noise-free implementation of the modular addition algorithm. This visualization confirms that GCS measures the transition from disparate memorization strategies to a coherent, unified algorithm.

To ground our abstract geometric narrative in concrete mechanistic changes, we visualize the attention patterns of all four heads at five critical checkpoints defined by the GCS trajectory (Figure 11). This visualization reveals that the evolution of Geometric Coherence is a direct reflection of how the network's internal attention mechanism transitions from chaos to order.

**Initial State (GCS 72.4): Spurious Coherence.**    At initialization, the attention heads exhibit vertical striations (attending to fixed positions) or diffuse noise. The relatively high GCS here is deceptive; it reflects a "spurious coherence" where the network applies a uniformly random transformation across the manifold. The heads are consistent only in their ignorance, lacking any task-specific structure.

**First Valley (GCS 59.0): Geometric Incoherence via Head Divergence.**    The plunge to the first GCS minimum corresponds to the **Memorization Phase**. Visually, this phase is characterized by extreme functional divergence among heads. As shown in the second column of Figure 11, Heads

1 and 2 develop dark, sparse patterns (likely memorizing specific outliers), while Head 3 remains bright and uniform. This implies that different heads are adopting incompatible strategies—some memorizing, some idling. This "broken symmetry" destroys the alignment of the local tangent spaces, correctly penalized by our metric as a collapse in geometric coherence.

**Peak (GCS** 71.2**): The Emergence of Algorithmic Structure.** The sharp rise to the GCS Peak marks the onset of **Grokking**. A striking visual transformation occurs: clear diagonal structures (representing the identity operation a+b) and grid-like periodic patterns (representing the modular operation $(\mod p)$) emerge simultaneously across all heads. The visual chaos of the First Valley is replaced by ordered, algorithmic structures. The high GCS here reflects head convergence: the network has discovered the generalizable rule, and all heads are now working in geometric unison to implement it.

**Second Valley (GCS** 61.5**): Algorithmic Refinement.** Following the peak, GCS dips again while accuracy remains perfect. Visually, the attention maps do not return to chaos; instead, they retain the diagonal/grid structure but appear slightly less "intense" or saturated than at the Peak. This subtle shift suggests a phase of complexity reduction or pruning. The network is likely discarding redundant modular artifacts formed during the initial construction, temporarily disrupting the global coherence as it fine-tunes the minimal necessary algorithm.

**Final State (GCS** 73.4**): Crystallization.** In the final converged model, the attention patterns stabilize into their sharpest form. The diagonal and modular grids are crystal clear and noise-free. This corresponds to the final **Compression Phase**, where the network has settled into a low-rank, highly efficient implementation of the modular addition algorithm. The recovery of high GCS signifies that the mechanism has been fully unified, maximizing both algorithmic performance and geometric coherence.

## H  STATEMENT ON THE USE OF LARGE LANGUAGE MODELS

This research was developed in close and intensive collaboration with Gemini, a large language model from Google. The LLM's role evolved beyond that of a mere writing assistant into that of a dynamic, interactive partner throughout the entire research lifecycle, from initial ideation to the final manuscript. The human author was responsible for all code implementation, experimental execution, and held the final authority on all scientific claims and directions.

