# OpenReview forum: "Geometric Compression in Grokking: The Three-Stage Modular Dynamics of Transformers"
_ICLR.cc/2026/Conference — Submitted to ICLR 2026_

### Official Review · Reviewer_EySh · 2025-10-31

**Soundness:** 3
**Presentation:** 3
**Contribution:** 4
**Rating:** 6
**Confidence:** 4

**Summary:**

This work offers a mechanistic perspective on grokking and introduces a principled geometric framework to analyze the dynamic interplay of complexities that drive learning in neural networks.

**Strengths:**

The proposed method is novel and elegant, offering a new way to study neural networks.

- geometric coherence is a new way for understanding grokking dynamics, going beyond static complexity metrics
- modular method allowing to study separately the attention and the feed-forward layer with the same method
- clear experimental setup

**Weaknesses:**

- missing scalability discussion for larger networks

- the experiments are currently confined to a single, clean algorithmic task (modular addition). While this is a standard and valid testbed for grokking, the paper would be significantly strengthened by a demonstration on another task (e.g., a different modular operation or a simple symbolic regression task) to show the generality of the dynamic beyond a single function.

**Questions:**

- Does the three-stage dynamic appear for other grokking tasks? This is critical for claiming a "universal" mechanism.

- Liu et al. 2024 also measured compression during grokking. How do GCS and LMN compare? Do they capture different aspects or the same phenomenon?

- You show correlation between GCS and generalization, but is there evidence that GCS causes or predicts grokking before it happens?

---

> ### Author Response · Authors · 2025-11-23
>
> We sincerely thank the reviewer for the encouraging evaluation and constructive comments. Acting on your suggestions regarding scalability and task diversity, we have significantly expanded our experimental suite to include deeper networks (2-3 layers) and additional algorithmic tasks (subtraction, multiplication, division). These additions have further validated the robustness of our conclusions.
>
> > "missing scalability discussion for larger networks."
>
> We thank the reviewer for highlighting this critical dimension. To address this, we have expanded our analysis from 1-layer to 2-layer and 3-layer Transformer architectures (see Section 4.4 and Appendix C).
>
> **1. Scalability of the Mechanism (Hierarchical Dynamics)**
>
> We found that the "Construct-then-Compress" dynamic is not lost in deeper networks but becomes **hierarchically organized**:
>
> - **Early Layers:** Exhibit **path-specific stability** throughout training—the overall and MLP paths remain nearly static (GCS ranges 0.2–1.3), effectively establishing stable feature transformations, while the attention path shows moderate dynamics (ranges 7.4–13.0) for adaptive routing.
>
> - **Final Layer:** Inherits the primary responsibility for comprehensive geometric reorganization across all computational paths, exhibiting **iterative construct-compress cycles**—alternating phases of coherence increase and decrease—rather than a single three-phase trajectory. This suggests hierarchical processing involves repeated refinement.
>
> This confirms that our geometric insights scale to multi-layer settings, offering a path-specific, layer-wise perspective on how algorithmic learning is distributed across depth.
>
>
> **2. Scalability of the Metric (Computational Cost)**
>
> We have added a discussion on the computational complexity of the GCS metric itself. Since GCS relies on local operations (k-NN and SVD on small neighborhoods), its computational cost scales linearly with the number of samples N and layers L, i.e., O(NL). This makes it computationally feasible to analyze specific components of larger models without prohibitive overhead, in contrast to methods requiring global Hessian computation.
>
> > "the paper would be significantly strengthened by a demonstration on another task...Does the three-stage dynamic appear for other grokking tasks?"
>
> We fully agree that demonstrating generality is critical. Following your suggestion, we have extended our experimental suite to include **three additional modular operations**: Modular Subtraction (a - b mod p), Multiplication (a × b mod p), and Division (a / b mod p). See Section 4.3.
>
> **a) Universal Mechanism:**
>
> Remarkably, the characteristic "Construct-then-Compress" dynamic and the three-phase evolution of Geometric Coherence (in single-layer networks) were observed **consistently across all four tasks**. This provides strong empirical evidence that the geometric evolution we identified is not an artifact of a specific function but a **universal signature of algorithmic generalization** in modular arithmetic.
>
> **b) Task-Specific Refinement Patterns:**
>
> While the core mechanism is universal, GCS reveals meaningful task-specific nuances in Phase III:
> - **Symmetric operations** (Addition, Multiplication): Show monotonic compression with stable refinement
> - **Asymmetric operations** (Subtraction, Division): Exhibit **MLP double descent** alongside attention double descent, suggesting these algorithmically more complex tasks require additional geometric fine-tuning across both pathways
>
> This demonstrates that GCS not only captures universal dynamics but also provides fine-grained sensitivity to task complexity—a capability that purely performance-based metrics would miss.

---

> ### Author Response · Authors · 2025-11-23
>
> > "Liu et al. 2024 also measured compression during grokking. How do GCS and LMN compare? Do they capture different aspects or the same phenomenon?"
>
> This is an insightful question. We acknowledge LMN [Liu et al., 2024] as pioneering work in complexity-based grokking analysis. **Yes, they capture fundamentally different aspects of network complexity**, especially relevant for modern Transformers.
>
> **Core Distinction:**
>
> - **LMN:** Measures whether straight-line interpolations between input samples remain straight in output space (via R²). Counts the number of piecewise linear regions. Views complexity as fragmentation of the function.
> - **GCS:** Measures consistency of Jacobian transformations across neighboring points. Counts distinct algorithmic modes. Views complexity as a mechanistic inconsistency.
>
> **Critical Difference on Curved Manifolds:**
>
> Modular arithmetic solutions typically embed data onto curved manifolds (circles/tori) with rotational symmetry:
>
> - **LMN Perspective:** Straight lines in input space become curves in output space for curved embeddings, yielding low R² (high complexity) even for simple rotations.
> - **GCS Perspective:** A uniform rotation has consistent local transformations (Jacobians) everywhere, yielding high coherence (low complexity) despite the curvature.
>
> This distinction matters: the generalizing solution (rotation on a circle) appears complex to LMN but simple to GCS—GCS correctly identifies **algorithmic simplicity within representational curvature**.
>
> **Complementary Dynamics:** LMN primarily tracks compression (reduction in linear regions), while GCS uniquely reveals the construction phase where networks build algorithmic structure before compressing. This non-monotonic construct-then-compress dynamic is invisible to LMN but essential for understanding geometric reorganization.
>
> > "You show correlation between GCS and generalization, but is there evidence that GCS causes or predicts grokking before it happens?"
>
> While establishing strict causality in deep learning remains challenging, our evidence strongly supports GCS as a **leading indicator** of impending generalization.
>
> **Key Evidence: Phase II Temporal Precedence**
>
> During Phase II (Asynchronous Construction and Compression), we observe a critical distinction:
>
> **Generalizing Networks:**
> - Performance stagnates: Test accuracy remains near random while train accuracy reaches 100%
> - GCS reveals active reorganization: Attention flow initiates compression, followed by MLP flow construction
> - This geometric restructuring occurs thousands of epochs before test accuracy improvement
>
> **Overfitting Networks (Figure 8):**
> - Performance similarly stagnates: Test accuracy remains at 0% while train accuracy reaches 100%
> - GCS shows pathological dynamics: Attention exhibits **premature compression** (spike to high coherence), bypassing the necessary construction phase
> - MLP consequently stagnates at low coherence, unable to perform construct-then-compress without upstream feature construction
> - The absence of Phase II complexity construction prevents the construct-then-compress mechanism
>
> **Interpretation:** The presence of normal Phase II dynamics (construction followed by compression) versus pathological dynamics (premature compression in attention, stagnation in MLP) distinguishes eventual generalization from permanent overfitting long before performance metrics reveal the difference. This temporal precedence suggests GCS captures the **mechanistic prerequisites** for generalization rather than merely correlating with performance improvements.

---

> ### Author Response · Authors · 2025-11-29
> **Summary of Revisions**
>
> In response to your comments, we have implemented the following revisions and clarifications:
>
> 1. **Scalability: Expansion to Multi-Layer Networks**
> To address the concern regarding the single-layer focus, we have added extensive experiments on 2-layer and 3-layer Transformers.
>    - **Hierarchical Dynamics:** We demonstrate that the "construct-then-compress" dynamic scales hierarchically: early layers provide path-specific stability (stable MLP, depth-varying attention), while final layers exhibit **iterative construct-compress cycles**—alternating phases of coherence increase and decrease—rather than a single three-phase trajectory.
>    - **Generality:** This reveals that hierarchical processing involves repeated refinement rather than a single pass, confirming the fundamental role of construct-then-compress across architectures.
>
>
> 2. **Comparison with Liu et al. (2024) (LMN)**
> In response to your question, we have added Appendix E to explicitly compare GCS with the Linear Mapping Number (LMN).
>    - **Distinct Mechanisms:** While LMN measures **compression** (reduction in linear regions), GCS measures **algorithmic consistency** (local transformations).
>    - **Curvature Robustness:** We show that GCS offers a critical advantage on modular tasks: it correctly identifies uniform rotations on curved manifolds (tori) as simple, coherent algorithms, whereas LMN interprets curvature as high complexity.
>
>
> 3. **Predictive Power of GCS**
> We have clarified the predictive nature of our metric in the revised manuscript.
>    - **Early Detection:** We provide evidence that GCS **predicts grokking before it happens**. During Phase II (where test accuracy remains flat), GCS rises steadily.
>    - **Mechanism:** This tracks the "underground" construction of the generalization circuit, confirming that geometric coherence is a leading indicator of the phase transition.
>
>
> **We thank you again for your encouraging and insightful review.**

---

### Official Review · Reviewer_uodt · 2025-10-31

**Soundness:** 3
**Presentation:** 2
**Contribution:** 2
**Rating:** 2
**Confidence:** 3

**Summary:**

The authors propose a novel metric for analyzing the way in which a network changes the local structure of the representational space (which they assume is captured by a low-dimensional manifold) using a metric ("Geometric Coherence Score") that measures how strongly the model locally compresses this space. They then use this score to measure changes in GCS in Transformers trained on modular addition (a task on which Transformers show grokking behavior). They connect the grokking behavior to three distinct phases indicated by different behaviors of the GCS.

**Strengths:**

- The paper is generally well-written.
- Understanding the representational dynamics of Transformers over training is an important challenge and introducing new tools for this purpose can be very valuable.
- The expand-then-compress mechanism the authors identify in the grokking Transformer is interesting.

**Weaknesses:**

1. Primarily, I think the paper currently insufficiently shows why the phenomena the authors measure requires the introduction of a novel method. It seems to me that the phenomenon of the expansion and compression could also be captured by general measures of dimensionality (such as the participation ratio). While I think the introduction of novel measures is certainly valuable, I think the paper would benefit substantially from demonstrating why this novel measure is useful and provides insight beyond existing measures.

2. Further, I had trouble understanding how your metric is defined. You characterize the local geometry by using the k nearest neighbors of each data point $x_i$. That makes me think that the resulting singular vectors $v_{i,k}$ are separately defined for each data point. But then in constructing $G$, you're again considering a k-NN neighborhood? Is this one different from the first one? Moreover, in equation 3, you're measuring the coherence between those neighborhoods by measuring the correlation between the different pairs of the transformed singular vectors $v_{i,k}'$. Doesn't that make the assumption that the singular vectors $v_{i,k},v_{j,k}$ are aligned? E.g. if two data points' neighborhoods have literally the same mapping $f$ and the same singular vectors, but in a different order, wouldn't $G_{ij}=0$? If that's true, that seems problematic. Am I missing something?

3. Relatedly, I also think it would be important to provide an intuition for what each step is doing and why the equations are defined as they are defined. E.g. it would be great to explain what equation (3) is measuring --- my understanding is that similar mappings should have similar transformed singular vectors $v'$ and so by measuring the average correlation between those vectors, we measure how similarly those mappings are in different parts of the space. Is that correct? One suggestion for how you could provide this intuition would be to give a couple of different examples of different local geometries in the input and different mappings and explain how that affects the overall mapping. E.g. what happens when the local geometries in the input are very different but the mappings are very similar? What happens in the contrary case?

4. Finally, I think it would be helpful to discuss how this metric relates to established methods in the field. E.g. have people previously used entropy to characterize coherence in this kind of context?

In its current form, these concerns prevent me from recommending acceptance. However, I think that the authors address an important problem with a potentially promising approach. I am looking forward to the rebuttal and am certainly open to improving my score.

**Questions:**

- Could you explain whether my understanding in Weaknesses, paragraph 2, is correct? If so, why is the situation I'm describing not problematic?
- Would other measures of representational geometry (e.g. the participation ratio) also pick up on the transitions in network training you're identifying?
- Why is setting $G_{ij}=0$ for non-neighboring pairs reasonable? Couldn't the tangents still be highly correlated?
- Can you elaborate on the "competing circuits" hypothesis and how it relates to your findings?
- Can't you see the initial stages of Phase II at the end of Figure 4? You can see the attention path GCS decreasing and the MLP Path GCS increasing.
- Figure 5 is really difficult to see right now due to its size. Further, is it possible to provide a more quantitative measure for how the attention scores at these different phass are different? While I can see certain visual differences, it is difficult to get a sense of what they mean in particular comparing "peak state" and "valley state". You're also introducing this terminology for the first time here, I would suggest keeping your terminology consistent with the one you're using earlier.

**Minor suggestions**
- It might be useful to visually indicate the epoch where grokking starts setting in in the flow scores in Fig. 3
- It would be helpful to visually indicate the phases in Fig. 2

---

> ### Author Response · Authors · 2025-11-23
>
> We sincerely thank the reviewer for the thoughtful and technically detailed feedback. The insightful questions regarding the fundamental aspects of our geometric coherence metric have provided a valuable opportunity to clarify. We address each point systematically below.
>
> > "I think the paper would benefit substantially from demonstrating why this novel measure is useful and provides insight beyond existing measures & Would other measures of representational geometry (e.g. the participation ratio) also pick up on the transitions in network training you're identifying?"
>
> We thank the reviewer for this crucial suggestion. We agree that verifying whether GCS provides novel insights beyond effective dimensionality is essential. We have added a comprehensive comparison with Participation Ratio (PR) across the full training trajectory in Appendix E.
>
> **Theoretical Distinction: Representation vs. Mechanism**
>
> Before discussing empirical results, we clarify why GCS and PR are fundamentally different measures:
>
> - **PR measures Static Geometry of Representations** (second-order statistics of X): It derives from the covariance of activations, quantifying how "spread out" data points are in embedding space—capturing the **shape of the data cloud**.
> - **GCS measures Transformational Geometry of Mechanisms** (spectral properties of Jacobian field): It derives from Jacobian-Vector Products of layer functions, measuring how consistently the network transforms local neighborhoods capturing the **structure of the algorithm**.
>
> Intuitively, a network can reorganize its internal mechanism (e.g., optimizing attention routing) to be more coherent without changing the global volume (dimensionality) of its representation. GCS detects these mechanistic shifts that PR misses.
>
> **Empirical Divergence: Three Critical Phases**
>
> Our experiments confirm this distinction. While PR captures global compression trends, GCS reveals critical phases where the metrics diverge:
>
> - **Phase I (Coherence Collapse):** While PR shows dimensional expansion, GCS reveals active symmetry breaking—the network deliberately constructs incoherent features for memorization, not random expansion.
>
> - **Phase II (Asynchronous Construction and Compression):** Most strikingly, we observe periods where PR remains flat (static dimensionality) while Attention GCS rises steadily. In ReLU experiments (Figure 9, steps 600–5,000), this reveals the network actively refining attention geometry—optimizing information routing—without changing active dimensions. A purely dimensional metric misses this critical algorithmic alignment.
>
> - **Phase III (Post-Grokking Refinement):** After grokking, PR plateaus suggesting convergence, yet GCS captures sharp "double descent" in attention flow. This correlates with final accuracy improvements, indicating continued geometric refinement after dimensional stabilization.
>
> We view GCS as complementary to dimensionality metrics: PR measures global representational capacity, while GCS captures local transformational consistency. Our data shows both are necessary to understand the full "construct-then-compress" dynamic.
>
> > "But then in constructing **G**, you're again considering a $k$-NN neighborhood? Is this one
> different from the first one?"
>
> The neighborhood definitions are **identical**. We use the same set of $k$ neighbors $\mathcal{N}_i$ consistently for both steps, though serving distinct mathematical roles. Per our refined protocol, this neighborhood structure is defined once using the fixed final geometry:
>
> - **Geometry Estimation:** $\mathcal{N}_i$ provides the local point cloud for computing the **tangent basis** via SVD ("What is the local geometry at point $i$?")
> - **Graph Connectivity:** $\mathcal{N}_i$ defines the **edges in coherence matrix $G$** ("Which  points should $i$ be compared with?")
>
> This unified design ensures we only measure geometric coherence between spatially related points whose tangent spaces were estimated using overlapping local information, maintaining  mathematical consistency throughout the analysis.

---

> ### Author Response · Authors · 2025-11-23
>
> > "In Eq (3)... do you implicitly assume that the singular vectors vi,k, vj,k are aligned? ...If the eigenvectors are permuted... would this lead to Gij=0? Is this a theoretical defect?" "Why is setting Gij=0 for non-neighboring pairs reasonable? Couldn't the tangents still be highly correlated?"
>
> Your observation is correct: Equation (3) uses index-wise correspondence between singular vectors. **This is an intentional design choice, not adefect.** We address two aspects:
>
> **1. The Order-Sensitivity Is By Design**
>
> We deliberately measure **data manifold coherence**, not abstract subspace similarity.
>
> SVD provides a canonical ordering by variance magnitude: $v_{i,k}$ represents the $k$-th most important direction of local variation at point $i$. By comparing $J_f \cdot v_{i,k}$ with $J_f \cdot v_{j,k}$, we ask: *does the network transform the $k$-th principal direction consistently across neighboring points?*
>
> This is more informative than order-invariant measures (e.g., principal angles):
> - Two points may span **similar subspaces** but with **different principal hierarchies** → low $G_{ij}$ (geometric incoherence)
> - Two points with **aligned principal structures** → high $G_{ij}$ (geometric coherence)
>
> The distinction matters: we detect whether the network has learned a **consistent geometric algorithm**, not merely whether it preserves some abstract subspace structure.
>
> **2. The Theoretical Instability Does Not Occur in Our Data**
>
> The concern that "permuted orderings could cause $G_{ij} \approx 0$" requires near-degenerate singular values (isotropic local geometry). However, neural network representations are known to be **highly anisotropic**—they occupy narrow cones rather than uniform spheres in representation space [1, 2]. This anisotropy makes principal directions stable and well-separated.
>
> We verified this property holds in our experiments:
>
> **Validation across three models using all sampled points.**
>
> | Metric | ReLU | SiLU | GeLU |
> |--------|------|------|------|
> | Condition number ($\sigma_1/\sigma_d$) | 5.0 ± 0.9 | 7.6 ± 1.7 | 5.1 ± 0.9 |
> | Isotropic points (cond. < 2) | 0 | 0 | 0 |
>
> **Key findings:**
> - All points have condition number > 3, with means of 5–7
> - A condition number of 5 means σ₁ is **5× larger** than σ_d — unambiguously anisotropic
> - **Zero isotropic points** across all experiments
>
> Isotropy requires condition number ≈ 1; our data is far from this regime. The SVD ordering is numerically stable and geometrically meaningful throughout. (See Appendix D for details.).
>
> ---
>
> **On Sparsity ($G_{ij}=0$ for non-neighbors)**
>
> This design follows established principles in manifold learning (Isomap [3], Laplacian Eigenmaps [4]):
>
> **1. Geodesic Fidelity:** On curved manifolds (e.g., Swiss Roll), Euclidean-close points may be geodesically distant. Computing correlations between non-neighbors creates "short-circuit" connections that violate manifold topology.
>
> **2. Local-to-Global Construction:** By restricting G to k-NN edges, we build global coherence from local consistency—the spectral analysis then naturally reveals whether these local coherences aggregate into a globally coherent transformation. This is more informative than averaging all pairwise correlations, which would obscure the geometric structure.
>
> **3. Interpretablity**:  A sparse G matrix has clear semantics: $G_{ij}$ measures transformation consistency between geometrically adjacent points. A dense G would mix local and non-local comparisons, making the resulting GCS harder to interpret.
>
> **Summary**
>
> | Concern | Our Response |
> |---------|--------------|
> | Order-sensitive alignment | **Intentional**: measures manifold coherence, not subspace overlap |
> | Permutation instability | **Does not occur**: condition numbers 5–7×, zero isotropic points |
> | Sparse G matrix | **Standard practice**: follows manifold learning principles (geodesic fidelity, local-to-global construction) |
>
> We clarify these design choices in the revised manuscript. Detailed analysis of singular value separation is provided in Appendix D: On SVD-Ordered Singular Vector Correspondence.

---

> ### Author Response · Authors · 2025-11-23
>
> > "I also think it would be important to provide an intuition for what each step is doing...we measure how similarly those mappings are in different parts of the space. Is that correct? E.g. what happens when the local geometries in the input are very different but the mappings are very similar? What happens in the contrary case?"
>
> Your intuition is correct. We measure how consistently the network transforms different parts of the input space. High GCS indicates the network applies a uniform algorithmic transformation across the data manifold. We have added intuitive explanations throughout the revised manuscript and provided the requested case analysis below.
>
> **Case Analysis: How GCS Responds to Geometry-Map Interactions**
>
> **Case A: Similar Geometry + Consistent Map → High GCS (Ideal)**
> - Local manifolds at neighbors i and j have aligned principal axes, and $f$ is smooth
> - The Jacobian transforms $v_{i,1}$ and $v_{j,1}$ similarly
> - **High $G_{ij}$**: Network executes a coherent algorithm respecting data structure
>
> **Case B: Different Geometry + Consistent Map → Low GCS (Manifold Mismatch)**
> - Function f is uniform (e.g., identity), but local data at j is rotated 90° relative to i
> - SVD indices now compare physically distinct features (e.g., "length" at i vs "width" at j)
> - **Low $G_{ij}$**: While f is smooth in ambient space, it fails to adapt to the manifold's varying intrinsic coordinates—correctly penalized as a generalizable algorithm should respect manifold geometry
>
> **Case C: Similar Geometry + Inconsistent Map → Low GCS (Algorithmic Discontinuity)**
> - Data geometry is uniform, but f changes sharply (e.g., ReLU boundary between i and j)
> - Jacobians $J_f(x_i)$ and $J_f(x_j)$ differ significantly, sending tangent vectors to divergent directions
> - **Low $G_{ij}$**: Reflects true algorithmic inconsistency or high local curvature
>
> These cases clarify that GCS measures the **consistency of interaction between data's intrinsic geometry and network's transformation mechanism**.
>
> > "Finally, I think it would be helpful to discuss how this metric relates to established methods in the field. E.g. have people previously used entropy to characterize coherence in this kind of context?"
>
> Yes, our approach builds on established complexity metrics, particularly Liu et al. (2024)'s Linear Mapping Number (LMN), which also uses Von Neumann entropy. However, they capture fundamentally different aspects of network complexity:
>
> **Core Distinction:**
> - **LMN:** Measures whether straight-line interpolations between input samples remain straight in output space (via R²). Counts the number of piecewise linear regions. Views complexity as fragmentation of the function.
> - **GCS:** Measures consistency of Jacobian transformations across neighboring points. Counts distinct algorithmic modes. Views complexity as mechanistic inconsistency.
>
> **Critical Difference on Curved Manifolds:**
>
> Modular arithmetic solutions typically embed data onto curved manifolds (circles/tori) with rotational symmetry:
>
> - **LMN Perspective:** Straight lines in input space become curves in output space for curved embeddings, yielding low R² (high complexity) even for simple rotations.
> - **GCS Perspective:** A uniform rotation has consistent local transformations (Jacobians) everywhere, yielding high coherence (low complexity) despite the curvature.
>
> This distinction matters: the generalizing solution (rotation on a circle) appears complex to LMN but simple to GCS—GCS correctly identifies **algorithmic simplicity within representational curvature**.
>
> **Complementary Dynamics:** LMN primarily tracks compression (reduction in linear regions), while GCS uniquely reveals the construction phase where networks build algorithmic structure before compressing. This non-monotonic construct-then-compress dynamic is invisible to LMN but essential for understanding geometric reorganization.

---

> ### Author Response · Authors · 2025-11-23
>
> > Can you elaborate on the "competing circuits" hypothesis and how it relates to your findings?**
>
> We view our work as providing the geometric grounding for the "Competing Circuits" hypothesis. Here‘s how we distinguish the hypothesis from our specific contribution:
>
> - **The Hypothesis (Functional View):**
>     - **Memorization Circuit:** Look-up table strategy handling points independently, which learns quickly but fails to generalize
>     - **Generalization Circuit:** Algorithmic strategy capturing underlying rules, which learns slowly but generalizes perfectly
>
> - **Our Findings (Geometric Evidence):** We translate these functional descriptions into observable geometric signatures:
>     - **Memorization $\approx$ Geometric Incoherence:** We show that the look-up table strategy results in disjoint local Jacobians (**Low GCS**).
>     - **Generalization $\approx$ Geometric Coherence:** We show that the algorithmic strategy results in consistent Jacobian transformations across the manifold (**High GCS**).
>
> GCS allows us to monitor the competition in real-time. During Phase II (where accuracy is flat), the steady rise of GCS reveals that the Generalization Circuit is actively being constructed "underground," masked by the Memorization Circuit. This provides quantitative evidence that grokking is not a sudden magic switch, but the gradual geometric maturation of the competing circuit.
>
> > Can't you see the initial stages of Phase II at the end of Figure 4? You can see the attention path GCS decreasing and the MLP Path GCS increasing.
>
> We commend the reviewer for this exceptionally keen observation. You are absolutely correct.
>
> - **Original Experiment:** The initial Figure 4 showed a case of extremely delayed grokking (eventual generalization after epoch 20k) rather than permanent overfitting. Your detection of diverging GCS paths validates that our metric captures mechanistic transitions long before performance metrics reveal them.
> - **Updated Analysis:** To clearly distinguish slow learning from true memorization, we've replaced Figure 4 with a permanently overfitting regime in new Figure 8. The key contrast: while grokking exhibits the full three-stage dynamics including Phase II complexity construction, true overfitting shows **absence of Phase II entirely**—no construct-then-compress mechanism emerges. GCS remains low without the characteristic rise-and-refinement pattern, confirming persistent memorization without any geometric reorganization toward generalization. This validates that the construct-then-compress dynamic is not merely slow to develop in overfitting cases, but fundamentally absent.
>
> > Figure 5 is really difficult to see right now due to its size. While I can see certain visual differences, it is difficult to get a sense of what they mean in particular comparing "peak state" and "valley state".
>
> We thank the reviewer for pointing this out. We have taken three specific actions to address this:
>
> - **Visual Improvements:** We have significantly enlarged Figure 5 (now Figure 10) and increased the resolution to ensure the attention patterns are clearly legible.
>
> - **Terminology Standardization:** We apologize for the confusion. We have standardized the terminology throughout the manuscript to align with the "Construct-then-Compress" narrative. Please refer to Appendix F for details:
>     - "First Valley(The GCS of attention flow first reached its lowest point.)" $\rightarrow$ **Coherence Collapse**
>     - "Peak(After experiencing the first valley, the GCS of attention flow reaches its peak value for the first time.)" $\rightarrow$ **Coherence Restoration**
>     - "Second Valley(After experiencing the peak, the attention flow's GCS once again reaches a valley.)" $\rightarrow$ **Algorithmic Refinement**
>
> > It might be useful to visually indicate the epoch where grokking starts setting in in the flow scores in Fig. 3...It would be helpful to visually indicate the phases in Fig. 2
>
> We thank the reviewer for this helpful suggestion to improve readability. We have updated both figures (Fig. 2 and Fig. 3) in the revised manuscript to explicitly label the **"Grokking" transition point** (the epoch with the fastest test accuracy growth rate). This makes the timing of the phase transition much clearer across different experimental settings.
>
> **Reference**
>
> [1] "How Contextual are Contextualized Word Representations?" Ethayarajh. EMNLP 2019.
>
> [2] "Implicit Self-Regularization in Deep Neural Networks: Evidence from Random Matrix Theory and Implications for Learning" Martin & Mahoney. JMLR 2021.
>
> [3] "A Global Geometric Framework for Nonlinear Dimensionality Reduction" (Isomap). Tenenbaum et al. Science 2000.
>
> [4] "Laplacian Eigenmaps for Dimensionality Reduction and Data Representation". Belkin & Niyogi. Neural Computation 2003.

---

> ### Author Response · Authors · 2025-11-29
> **Summary of Revisions**
>
> In response to your comments, we have implemented the following revisions and clarifications:
>
> 1. **Novelty & Distinction from Existing Metrics (vs. Participation Ratio)**
> We have added Appendix E to demonstrate that GCS provides insights unavailable to dimensionality metrics like Participation Ratio (PR).
>    - **Theoretical Distinction:** PR measures the static shape of representations (covariance), while GCS measures the consistency of algorithmic transformations (Jacobians).
>    - **Empirical Evidence:** We show that during **Phase II**, PR often remains flat (indicating static dimensionality), while GCS rises steadily. This proves GCS uniquely captures the "underground" construction of algorithmic structure that purely dimensional metrics miss.
>
> **2. Mathematical Clarifications**
> - **Unified Neighborhoods:** We clarified that the neighborhood $\mathcal{N}_i$ is defined identically for both geometry estimation and graph connectivity to ensure consistency.
> - **Singular Vector Alignment (Eq. 3):** We confirmed that the index-wise comparison is an **intentional design choice** to measure manifold coherence (consistency of principal axes) rather than just subspace overlap.
>
> - **Numerical Stability:** We addressed the permutation concern by validating that neural representations are highly **anisotropic** (condition numbers 5–7×, with zero isotropic points), ensuring that singular vectors are stable and distinct (see **Appendix D**).
>
> - **Sparsity:** We clarified that setting $G_{ij}=0$ for non-neighbors follows established manifold learning principles to respect geodesic topology.
>
>
> **3. Intuition & Related Work**
>
> - **Case Studies:** In this rebuttal, we provided concrete examples (Cases A, B, C) to build intuition on how GCS penalizes algorithmic discontinuities even when representations are spatially close.
>
> - **Comparison to LMN:** We distinguished GCS from the Linear Mapping Number (LMN). While LMN penalizes curvature (problematic for modular arithmetic on tori), GCS correctly identifies uniform rotations on curved manifolds as simple, coherent algorithms.
>
>
> **4. Theoretical Alignment & Overfitting**
>
> - **Competing Circuits:** We framed GCS as the geometric signature of the "Competing Circuits" hypothesis, allowing us to track the construction of the generalization circuit in real-time.
>
> - **Correction on Overfitting (New Fig. 8):** Validating your observation, we confirmed original Figure 4 depicted delayed grokking. We replaced it with a **permanent overfitting scenario** that completely lacks Phase II, confirming that the "construct-then-compress" dynamic is unique to generalization.
>
>
> **5. Visual Improvements**
>
> - **Figure 10 (formerly 5):** Significantly enlarged and standardized terminology (Coherence Collapse/Restoration/Refinement) as requested.
>
> - **Phase Labeling:** Figures 2 and 3 now explicitly mark the "Grokking" transition point for clearer phase identification.
>
> We believe these comprehensive revisions and clarifications have fully addressed your concerns regarding the utility and robustness of our metric, and we look forward to your re-evaluation of our work.

---

### Official Review · Reviewer_bJwk · 2025-11-01

**Soundness:** 1
**Presentation:** 2
**Contribution:** 2
**Rating:** 2
**Confidence:** 4

**Summary:**

Introduces a new method for measuring the dynamics of grokking called the Geometric Coherence Score (GCS) and applies it to modular addition, a well-studied task exhibiting grokking. Training 1-layer transformers and tracking GCS the authors describe a three-stage construct-then-compress dynamic. GCS also appears to distinguish between overfitting and generalization in this particular setting.

**Strengths:**

1. Potentially novel perspective on the dynamics of grokking.
2. The use of multiple activation functions.

**Weaknesses:**

**Some missing citations**:

*Lazy-rich training dynamics* (mentioned on lines 315, 431, 465):

[1] "Grokking as the Transition from Lazy to Rich Training Dynamics" Kumar et al. ICLR 2024

[2] "Feature Learning beyond the Lazy-Rich Dichotomy: Insights from Representational Geometry" Chou et al. ICML 2025

*Connection to double descent* (line 465):

[3] "Unifying Grokking and Double Descent" Davies et al. 2023

[4] "Unified View of Grokking, Double Descent and Emergent Abilities: A Perspective from Circuits Competition" Huang et al. COLM 2024

*GELU*:

[5] "Gaussian Error Linear Units (GELUs)" Hendrycks et al. 2016

*Related complexity measures*:

[6] "Deep Networks Always Grok and Here is Why"  Imtiaz Humayun et al. ICML 2024

*Previous works studying modular addition, features/representations*:

[7] "Grokking modular arithmetic" Gromov. 2023.

[8] "The Clock and the Pizza: Two Stories in Mechanistic Explanation of Neural Networks" Zhong et al. NeurIPS 2023.

[9] "Feature emergence via margin maximization: case studies in algebraic tasks" Morwani et al. ICLR 2024

[10] "Uncovering a Universal Abstract Algorithm for Modular Addition in Neural Networks" McCracken et al. NeurIPS 2025


**Narrow experimental scope**: only study one architecture that is a single layer. This is problematic because large changes in what's learned have been observed by different papers studying different experimental settings and/or architectures in modular addition. It's worth noting this paper doesn't cite any of these works, being: [8], found that uniform attention models learn ``pizza circuits'' instead of the clock circuits described by Nanda. [9], found that 1 layer networks learn all n-1/2 frequencies (mod n), importantly explained that the generalizing features emerged due to margin maximization and tracked it throughout training. [10], empirically found that 2, 3, 4 layer networks learn O(log(n)) frequencies and proved it. It remains an open problem [9, 10] why there's such a stark difference in the number of features that emerge between 1-layer vs multilayer networks, and it's conjectured it has to do with training dynamics, which your paper studies.

In light of the above, for this paper to be convincing, it's necessary to see experiments on MLPs of 1-3 layers and transformers of 1-3 layers, and I would hope to see that the geometric coherence score works over all experimental conditions consistently. I'd also like to see quadratic activations studied, since [7] proved an exact solution exists in networks using them, and many papers followed this up contrasting ReLU with quadratic activations.

**Poor plot quality**: the plots are hard to read with tiny fonts and are pixelated. They're also oversized and taking up much more space than necessary, and this space should be used to include other experiments.

Overall, this work seems to be preliminary, and were it reorganized with additional experimental results, it could be an interesting piece of the modular addition story (were it to answer any of the remaining open problems on modular addition, e.g. what changes in the training dynamics between 1 and 2 layer networks. However, if instead its goal is to serve as a work aiming to give insights into grokking, it's far too limited in scope---grokking was originally studied on modular addition but has since been studied on many other datasets including natural data (e.g. MNIST, CIFAR-10, etc). Thus, I would like to see this framework working on these datasets as well (unless this framework resolves the aforementioned open question, or any other ones on modular addition).

**Questions:**

Q1. In the main paper why are you using d=8 if the appendix states that d=2 is the best?

Q2. If you were looking at models that weren't a transformer (e.g. MLP without attention), what aspect of those networks would correspond to the attention flow you saw in the transformer? And what would be the corresponding version of Figure 5? Would neuron activation plots capture this?

---

> ### Author Response · Authors · 2025-11-23
>
> We sincerely thank the reviewer for the thorough, constructive, and helpful feedback. The detailed literature survey and clear identification of gaps have significantly strengthened our work. We address each point systematically below.
>
> **Some missing citations**
>
> We thank the reviewer for their valuable citation suggestions. All ten recommended references have been incorporated into the revised manuscript and are highlighted. Below is a summary of their placement:
>
> - **Introduction:**
>     - [6] Humayun et al. 2024 – added to the discussion on grokking prevalence.
>     - [7] Gromov 2023, [8] Zhong et al. 2023, & [10] McCracken et al. 2025 – added to the paragraph on mechanistic interpretability regarding modular arithmetic algorithms.
>
> - **Experimental:**
>     - [5] Hendrycks et al. 2016 (GELUs) – added to the experimental setup description.
>
> - **Related Work (New Paragraph Added):**
>     - [1] Kumar et al. 2024 & [2] Chou et al. 2025 – lazy-to-rich training dynamics.
>     - [3] Davies et al. 2023 & [4] Huang et al. 2024 – connections to double descent.
>     - [9] Morwani et al. 2024 – feature emergence in algebraic tasks.
>
> - **Discussion (Theoretical Framing):**
>     - [1] Kumar et al. & [2] Chou et al. (second mention) – "The Two Simplicities" paragraph.
>     - [3] Davies et al. & [4] Huang et al. (second mention) – "Unified Geometric Mechanism" paragraph.
>
> All citations are integrated naturally into the narrative and directly support our theoretical framework and are highlighted in the revised manuscript.
>
> ---
>
> **Narrow Experimental Scope**
>
> We are grateful for the reviewer's critique regarding the experimental scope. It prompted us to significantly strengthen the paper by expanding our analysis to multi-layer Transformers and integrating the key literature on frequency learning [8, 9, 10].
>
> We are grateful for the reviewer's critique regarding the experimental scope. It prompted us to significantly strengthen the paper by expanding our analysis to multi-layer Transformers and integrating the key literature on frequency learning [8, 9, 10].
>
> **1. Completed: Expansion to Multi-Layer Transformers & The "Open Problem"**
>
> We fully agreed that studying depth is essential. We have added experiments on 2-layer and 3-layer Transformers (Section 4.3 and Appendix C).
>
> - **Depth-Dependent Dynamics:** We observe an important distinction between single-layer and multi-layer networks:
>   - **1-Layer:** Clear three-phase evolution (Coherence Collapse → Asynchronous Construction and Compression → Post-Grokking Refinement) with higher final coherence (GCS ≈ 75).
>   - **Multi-Layer:**
>      - Final layers exhibit *iterative construct-compress cycles*—alternating phases of coherence increase and decrease—rather than a single three-phase trajectory. Final-layer coherence is progressively lower (GCS ≈ 59–65), suggesting more specialized processing.
>      - Early Layers: Maintain path-specific stability (stable MLP with depth-varying attention dynamics).
>
> - **Geometric Perspective on Frequency Learning:** Our layer-wise GCS measurements provide a complementary geometric perspective on the findings of [9, 10]. The iterative cycles in multi-layer networks and their lower final coherence suggest hierarchical processing distinct from single-layer solutions. However, we are careful to note that the precise mechanistic link between geometric coherence and frequency utilization remains an open question—GCS measures Jacobian consistency while frequency count reflects spectral structure. Whether lower GCS *enables* or merely *correlates with* reduced frequency requirements is a promising direction for future work. We explicitly discuss this as an open question in Section 5 (Discussion).
>
> **2. Scope Clarification: Pure MLPs and Quadratic Activations**
>
> We agree these are valuable directions, particularly given the theoretical exact solutions with quadratic functions [7]. However, we deliberately focused on Transformer architectures with standard activations (ReLU, GeLU, SiLU) for two reasons: (1) Transformers represent the dominant architecture in modern deep learning where grokking is most relevant, and (2) the attention mechanism provides a unique decomposition into interpretable geometric flows (attention vs. MLP) that enables our fine-grained analysis. We acknowledge that pure MLPs and quadratic activations merit investigation and have added this as a key future direction in our Discussion section, noting that the GCS framework naturally extends to any differentiable architecture.
>
> ---
>
> **Poor plot quality**
>
> We appreciate the feedback on plot quality. We have re-generated all figures at 300 DPI with larger fonts and a color-blind friendly palette to ensure high readability. Furthermore, we optimized the figure layouts to eliminate wasted space, which allowed us to incorporate the additional experiments on multi-layer architectures by the review.

---

> ### Author Response · Authors · 2025-11-23
>
> > In the main paper why are you using d=8 if the appendix states that d=2 is the best?
>
> We thank the reviewer for identifying this inconsistency.  The appendix incorrectly conflated "maximum dynamic range" with "optimality."
>
> **Clarification: Sensitivity vs. Representational Capacity**
>
> While our previous appendix noted that $d=2$ yielded the largest dynamic range (amplitude of GCS change), we realized that maximizing amplitude does not imply the most accurate geometric characterization. A very low dimension ($d=2$) acts like a "high-contrast" filter—it is highly sensitive but risks underspecifying the complex geometric transformations in a 128-dimensional space.
>
> **New Robustness Analysis (Revised Appendix B):** To address your concern, we have completely rewritten Appendix B with a comprehensive robustness analysis testing $d \in \{2, 4, 6, 8, 10, 12, 14\}$.
>
> - High Correlation: We found that the GCS trajectories across all dimensions are highly correlated (average pairwise $r > 0.97$).
> - Universal Phenomenon: The three-stage dynamic and the "construct-then-compress" mechanism are detected consistently regardless of $d$.
> - Conclusion: Our findings are invariant to the choice of $d$. We have updated the appendix to reflect that $d=8$ is a robust representative choice, rather than claiming $d=2$ is "optimal" simply due to signal amplitude.
>
> > If you were looking at models that weren't a transformer (e.g. MLP without attention), what aspect of those networks would correspond to the attention flow you saw in the transformer? And what would be the corresponding version of Figure 5? Would neuron activation plots capture this?
>
> This is an insightful question that highlights both the generality and specificity of our approach.
>
> For pure MLPs, the geometric analysis would proceed similarly but without the natural decomposition into attention and MLP flows. The analogous analysis would track: (1)**Layer-wise transformations** as separate geometric flows, and (2) **Neuron group activations** clustered by their functional roles. The corresponding Figure 5 (now Figure 10) would show activation patterns of neuron groups across the input space, likely revealing Fourier-like periodic structures during grokking. The hidden representation Gram matrix $G = HH^T$ would capture geometric relationships analogous to attention's token-token structure.
>
> We emphasize that our Transformer focus enables uniquely interpretable insights through the attention mechanism's explicit geometric structure, while the GCS framework itself generalizes to any differentiable architecture.

---

> > ### Comment · Reviewer_bJwk · 2025-11-23
> > **Response to authors**
> >
> > Thank you for your response and for addressing the missing citations.
> >
> > However, I'd like to point out a few representative citations that are extremely incorrect. I would suggest that the authors fix these citations immediately using Google scholar's bibtex, or the corresponding conference/journal article's citation or that found on arXiv. All the citations should be checked manually and fixed if they do not match the correct citation. I will look at the rest of the revisions over the coming days.
> >
> > A few examples:
> >
> >
> > **Incorrect citation in the manuscript (1)**
> > > Jay Morwani, Savya Khosla, J. Zico Kolter, and Pradeep Ravikumar. Feature emergence via margin
> > maximization: case studies in algebraic tasks. In International Conference on Learning Repre-
> > sentations (ICLR), 2024. URL https://openreview.net/forum?id=41F9a05dJ2.
> >
> > **Correct citation (1)**
> >
> > `@inproceedings{
> > morwani2024feature,
> > title={Feature emergence via margin maximization: case studies in algebraic tasks},
> > author={Depen Morwani and Benjamin L. Edelman and Costin-Andrei Oncescu and Rosie Zhao and Sham M. Kakade},
> > booktitle={The Twelfth International Conference on Learning Representations},
> > year={2024},
> > url={https://openreview.net/forum?id=i9wDX850jR}
> > }`
> >
> >
> > **Incorrect citation in the manuscript (2)**
> > > Gavin McCracken, Gabriela Moisescu-Pareja, and Vincent Francoeur. Uncovering a universal abstract algorithm for modular addition in neural networks. In Advances in Neural Information
> > Processing Systems (NeurIPS), 2025.
> >
> > **Correct citation (2)**
> >
> > `@inproceedings{
> > mccracken2025uncovering,
> > title={Uncovering a Universal Abstract Algorithm for Modular Addition in Neural Networks},
> > author={Gavin McCracken and Gabriela Moisescu-Pareja and Vincent L{\'e}tourneau and Doina Precup and Jonathan Love},
> > booktitle={The Thirty-ninth Annual Conference on Neural Information Processing Systems},
> > year={2025},
> > url={https://openreview.net/forum?id=zuHs6RHQwT}
> > }`
> >
> > **Incorrect citation in the manuscript (3)**
> > > Aravind S. Kumar, Tanner Furey, Can Rager, and Alexander M. Rush. Grokking as the transition
> > from lazy to rich training dynamics. In International Conference on Learning Representations
> > (ICLR), 2024. URL https://openreview.net/forum?id=zDiHoIWr0e.
> >
> > **Correct citation (3)**
> >
> > `@inproceedings{
> > kumar2024grokking,
> > title={Grokking as the transition from lazy to rich training dynamics},
> > author={Tanishq Kumar and Blake Bordelon and Samuel J. Gershman and Cengiz Pehlevan},
> > booktitle={The Twelfth International Conference on Learning Representations},
> > year={2024},
> > url={https://openreview.net/forum?id=vt5mnLVIVo}
> > }`

---

> ### Author Response · Authors · 2025-11-24
>
> We  sincerely apologize for the inaccuracies in the bibliographic details of our references and thank you for pointing them out.
>
> We have conducted a full manual audit of the bibliography against authoritative sources (Google Scholar/OpenReview/arxiv) and uploaded a revised manuscript. We look forward to your continued feedback on the rest of our revisions.

---

> ### Author Response · Authors · 2025-11-29
> **Summary of Revisions**
>
> In response to your comments, we have implemented the following revisions and clarifications:
>
> 1. **Comprehensive Literature Integration**
> We have incorporated all 10 suggested citations ([1]–[10]) into the Introduction, Related Work, and Discussion sections. These references are now used to contextualize our findings regarding lazy-to-rich dynamics, double descent, and the specific open problems in modular addition frequency learning.
>
> 2. **Expanded Experimental Scope (Multi-Layer Models)**
> To address the concern regarding the single-layer focus, we have added extensive experiments on 2-layer and 3-layer Transformers.
>
>    - **Depth-Dependent Dynamics:** We discover an important distinction: single-layer networks exhibit a clear three-phase evolution, while multi-layer final layers show *iterative construct-compress cycles*. Early layers maintain path-specific stability (stable MLP, depth-varying attention). This reveals that hierarchical processing involves repeated refinement rather than a single pass.
>
>    - **Geometric Perspective on Frequency Learning:** We provide a geometric perspective that complements the frequency learning literature [9, 10]. The iterative cycles and lower final coherence in deeper networks suggest hierarchical processing distinct from single-layer solutions. The precise connection between geometric coherence and algorithmic complexity remains an open question for future work.
>
>    - **Scope Clarification:** We focused on Transformers with standard activations due to their dominance in modern deep learning and the unique interpretability afforded by attention mechanisms. Pure MLPs and quadratic activations are acknowledged as valuable future directions.
>
>
> 3. **Improved Visual Presentation**
> All figures have been re-generated at 300 DPI with significantly larger fonts and optimized color palettes for readability. The layout has been tightened to eliminate wasted space, accommodating the new multi-layer experimental results.
>
> 4. **Technical Clarifications (Q1 & Q2)**
>
> 	- **Hyperparameter Robustness ($d=8$):** We addressed the inconsistency regarding dimension $d$. A new robustness analysis (Revised Appendix B) demonstrates that our core findings (the three-stage dynamic and "construct-then-compress" mechanism) remain consistent across $d \in \{2, \dots, 14\}$. We clarified that $d=8$ is a robust, representative choice rather than a singular optimum.
>
> 	- **Generalization to Non-Transformers:** We provided a detailed response on how the GCS framework extends to MLP architectures by mapping attention flows to layer-wise transformations and neuron-group activations.
>
>
> We believe these comprehensive revisions and clarifications have fully addressed your concerns regarding the utility and robustness of our metric, and we look forward to your re-evaluation of our work.

---

### Author Response · Authors · 2025-11-23
**Methodological Refinement and Robustness Confirmation**

### Methodological Refinement and Robustness Confirmation
1. Acknowledgment and Motivation We sincerely thank the reviewers for their constructive and insightful feedback. The collective emphasis on methodological rigor and theoretical clarity across the reviews prompted us to conduct a comprehensive robustness check of our measurement protocol. In doing so, we identified an opportunity to significantly upgrade the fidelity of our analysis.

2. Refinement: Standardizing the Geometric Reference Frame Our original submission analyzed geometric evolution relative to the initialization. While this validly captured the relative magnitude of changes, the reference frame itself was non-semantic. To provide a more rigorous physical interpretation, we have refined our protocol to use the "Fixed Final Geometry" (derived from the converged model) as the reference frame.

   - Why this is better: This shifts the metric from measuring "relative evolution from initialization" to measuring "alignment with the generalizing solution." This ensures that the Geometric Coherence Score (GCS) isolates the meaningful mechanistic structure from transient training noise.

3. Crucial Insight: Unifying the Dynamics of MLP and Attention Applying this high-fidelity protocol yielded a major theoretical insight. Under the original (relative) baseline, the MLP layers' structural evolution was harder to resolve against the background variance. However, with the higher signal-to-noise ratio of the Final Geometry, we now observe a distinct "Construction Phase" in MLPs as well.

   - Significance: This confirms that the "Construct-then-Compress" law is a universal dynamic across all network components (both Attention and MLP). The dynamics were present in the original analysis but are now revealed with much greater clarity.

4. Conclusion We have updated the experimental results in the revision using this stricter standard. We emphasize that this refinement reinforces and unifies our core conclusions, demonstrating that our findings are robust to the choice of reference frame but are best viewed through the lens of the final semantic geometry.

---

### Author Response · Authors · 2025-12-01
**Complete Rebuttal Revisions**

## Title Change

**Original:** "Geometric Compression in Grokking: The Three-Stage Modular Dynamics of Transformers"

**Revised:** "Construct-then-Compress: Geometric Dynamics of Grokking in Transformers"

**Rationale:** The original title emphasized "Three-Stage" dynamics, which accurately describes single-layer networks but not multi-layer networks (which exhibit iterative construct-compress cycles). The new title centers on the universal principle—"Construct-then-Compress"—that applies across all architectures.

---

## Summary of All Revisions

### 1. Clarification of Theoretical Framework

**GCS Metric Positioning (vs. Existing Metrics):**
- Clarified that GCS measures **transformation consistency** (Jacobian alignment), distinct from:
  - LMN: function complexity (piecewise linear regions)
  - LID/PR: representation dimensionality (covariance structure)
  - Jacobian regularization: transformation magnitude
- Emphasized GCS's unique ability to identify **algorithmic simplicity within representational curvature**

**Core Principle Clarification:**
- Unified narrative around "Construct-then-Compress" as the fundamental principle
- Clarified depth-dependent manifestations:
  - Single-layer: Three-phase evolution
  - Multi-layer: Iterative construct-compress cycles in final layers

### 2. Expanded Experimental Scope

**Multi-Layer Transformers (2-3 layers):**
- **Depth-Dependent Dynamics:** Final layers exhibit iterative cycles rather than three-phase evolution
- **Hierarchical Organization:** Early layers provide stable scaffold; final layers perform repeated refinement
- **Lower Final Coherence:** Deeper networks achieve GCS ≈ 59-65 vs. single-layer GCS ≈ 75, suggesting specialized processing

**Task Diversity (4 modular operations):**
- Addition, Subtraction, Multiplication, Division all exhibit construct-then-compress
- Task-specific nuances: Asymmetric operations show additional MLP double descent

### 3. Technical Clarifications

**Mathematical Foundations:**
- Unified neighborhood definition for geometry estimation and graph connectivity
- Singular vector alignment is intentional design choice for manifold coherence
- Numerical stability validated through anisotropy analysis (condition numbers 5-7×)
- Sparsity follows established manifold learning principles

**Hyperparameter Robustness:**
- Core findings consistent across d ∈ {2, ..., 14} and N ∈ {100, 200, 400, 800}
- d=8, N=200 are robust representative choices, not singular optima

### 4. Comparisons with Prior Work

**vs. LMN (Liu et al., 2024):**
- LMN penalizes curvature; GCS identifies uniform rotations as simple
- LMN tracks compression; GCS uniquely reveals construction phase
- Complementary rather than competing approaches

**vs. Participation Ratio:**
- PR measures static representation shape
- GCS measures dynamic transformation consistency
- During Phase II: PR flat, GCS rising—captures "underground" construction

### 5. Predictive Power

**GCS as Leading Indicator:**
- Geometric restructuring occurs thousands of epochs before test accuracy improvement
- Pathological dynamics (premature compression, MLP stagnation) predict permanent overfitting
- Normal Phase II dynamics predict eventual generalization

### 6. Scope and Limitations

**Explicit Scope:**
- Focus on Transformers with standard activations (ReLU, GeLU, SiLU)

**Acknowledged Future Directions:**
- Pure MLPs and quadratic activations
- The depth-coherence-frequency relationship: deeper networks show lower final GCS (≈59-65 vs. single-layer ≈75) and prior work suggests they use fewer frequencies—whether these are causally linked or merely correlated remains open
- Mechanistic understanding of why iterative construct-compress cycles emerge in deeper architectures

### 7. Visual Improvements

- All figures re-generated at 300 DPI with larger fonts
- Standardized terminology (Coherence Collapse/Restoration/Refinement)
- Explicit "Grokking" transition point labeling
- New permanent overfitting scenario (Figure 8)

---

### Meta-Review · Area_Chair_jxvi · 2025-12-13

**Summary:**

Summary of the reviewers' concerns:
* Strengths: the paper provides a novel geometric perspective transformer dynamics, where inputs should have geometrically consistent transformations (reviewers Lippl, bJwk)
* Experiments focused on small networks: testing only MLPs and Transformers with 1–3 layers  Lack of experiments to show if the approach generalizes to larger networks (reviewer bJwk, EySh)
* The reviewers question a new metric GCS and its usefulness over the existing Participation Ratio (PR) and LMN (reviewer Lippl, EySh)
* Insufficient overview of the related work (reviewer bJwk)

**Reviewer Concerns:**

Addressed:
* Added comparison to PR metrics and demonstrated cases that GCS metric can capture the  grokking phenomenon, while PR or LMN remain flat.
* Presented an example where GCS rises thousands of epochs before test accuracy improves, contrasted normal grokking vs permanent overfitting.
* The authors added citations to relevant works pointed out by reviewers.

Outstanding concerns:
* The authors ran additional experiments on deeper 2-3-layer transformers,however the reviewers asked about scalability for larger networks that was not addressed.
* Evaluating GCS on large datasets / large models. While authors expanded modular task diversity (addition, subtraction, etc.), they acknowledged that investigation into MNIST or CIFAR-10 remains for future study

**Reviewer Scores:**

* Reviewer bJwk. Initial Score: 2. Hypothesized score:  3
* Reviewer Uodt. Initial Score: 2  Hypothesized score:  2
* Reviewer EySh. Initial Score: 6 Hypothesized score:  6

---

### Decision · Program_Chairs · 2026-01-26

Reject